# Yeast9: a consensus genome-scale metabolic model for *S. cerevisiae* curated by the community

Chengyu Zhang [1,2], Benjamín J Sánchez [3,4], Feiran Li [5], Cheng Wei Quan Eiden[6], William T Scott [7,8], Ulf W Liebal[9], Lars M Blank [9], Hendrik G Mengers[9], Mihail Anton [10], Albert Tafur Rangel [3,11], Sebastián N Mendoza [12,13], Lixin Zhang[2], Jens Nielsen[11,14], Hongzhong Lu [1✉] & Eduard J Kerkhoven [3,15✉]

## Abstract

**Genome-scale metabolic models (GEMs) can facilitate metabolism-focused multi-omics integrative analysis. Since Yeast8, the yeast-GEM of *Saccharomyces cerevisiae*, published in 2019, has been continuously updated by the community. This has increased the quality and scope of the model, culminating now in Yeast9. To evaluate its predictive performance, we generated 163 condition-specific GEMs constrained by single-cell transcriptomics from osmotic pressure or reference conditions. Comparative flux analysis showed that yeast adapting to high osmotic pressure benefits from upregulating fluxes through central carbon metabolism. Furthermore, combining Yeast9 with proteomics revealed metabolic rewiring underlying its preference for nitrogen sources. Lastly, we created strain-specific GEMs (ssGEMs) constrained by transcriptomics for 1229 mutant strains. Well able to predict the strains' growth rates, fluxomics from those large-scale ssGEMs outperformed transcriptomics in predicting functional categories for all studied genes in machine learning models. Based on those findings we anticipate that Yeast9 will continue to empower systems biology studies of yeast metabolism.**

**Keywords** *Saccharomyces cerevisiae*; Genome-scale Metabolic Models; Machine Learning; Multi-omics Integration
**Subject Categories** Computational Biology; Metabolism

## Introduction

*Saccharomyces cerevisiae*, a widely used model organism in eukaryotic studies, was the first eukaryote whose genome was thoroughly sequenced and annotated (Goffeau et al, 1996). *S. cerevisiae* has long been used to study genetic interactions (Costanzo et al, 2016; Tong et al, 2004), build cell factories for the production of high-value-added compounds (Paddon et al, 2013; Chen et al, 2020; Ro et al, 2006), and comprehend eukaryotic metabolism due to its clear genetic background, abundant gene annotation and genetic tractability (DiCarlo et al, 2013; Jacobus et al, 2022; Mans et al, 2015). Having amassed extensive knowledge of the metabolism and physiology of yeast such as *S. cerevisiae*, the genome-scale metabolic models (GEMs) of this organism have undergone 20 years of iterative refinement and enhancement including Yeast8 (Lu et al, 2019), Yeast7 (Aung et al, 2013), Yeast6 (Herrgård et al, 2008), iND750 (Duarte et al, 2004), iLL672 (Kuepfer et al, 2005) since the initial publication of the first-generation model, iFF708, in 2003 (Förster et al, 2003). The establishment and maturation of yeast-GEMs have laid a strong foundation for the emergence of a model ecosystem centered on *S. cerevisiae*, including ecYeast, etcYeast, and pcYeast, collectively enabling a variety of system and synthetic biology investigations concerning *S. cerevisiae* (Lu et al, 2019, 2022). For instance, leveraging the enforced objective flux (FSEOF) algorithm (Choi et al, 2010), Yeast8 and ecYeast8 have been used to obtain a 70-fold improvement in heme production (Ishchuk et al, 2022). Kinetic models derived from yeast-GEM possess the ability to discern species-specific behavior (Hu et al, 2023; Henriques et al, 2021; Scott et al, 2023).

The ability by which yeast, like any other cellular biocatalyst, can overproduce desirable chemicals and secrete commercial proteins

[1]State Key Laboratory of Microbial Metabolism, School of Life Sciences and Biotechnology, Shanghai Jiao Tong University, 200240 Shanghai, China. [2]State Key Laboratory of Bioreactor Engineering, and School of Biotechnology, East China University of Science and Technology (ECUST), 200237 Shanghai, China. [3]The Novo Nordisk Foundation Center for Biosustainability, Technical University of Denmark, DK-2800 Kgs Lyngby, Denmark. [4]Department of Biotechnology and Biomedicine, Technical University of Denmark, DK-2800 Kgs Lyngby, Denmark. [5]Institute of Biopharmaceutical and Health Engineering, Tsinghua Shenzhen International Graduate School, Tsinghua University, Shenzhen 518055, China. [6]School of Chemistry, Chemical Engineering and Biotechnology, Nanyang Technological University, 62 Nanyang Drive, Singapore 637459, Singapore. [7]UNLOCK, Wageningen University & Research, Wageningen, The Netherlands. [8]Laboratory of Systems and Synthetic Biology, Wageningen University & Research, Wageningen, The Netherlands. [9]Institute of Applied Microbiology - iAMB, Aachen Biology and Biotechnology - ABBt, RWTH Aachen University, 52074 Aachen, Germany. [10]Department of Life Sciences, National Bioinformatics Infrastructure Sweden, Science for Life Laboratory, Chalmers University of Technology, Gothenburg SE412 58, Sweden. [11]Department of Life Sciences, Chalmers University of Technology, Gothenburg SE412 96, Sweden. [12]Center for Mathematical Modeling, University of Chile, Santiago, Chile. [13]Systems Biology Lab, Vrije Universiteit Amsterdam, Amsterdam, The Netherlands. [14]BioInnovation Institute, Ole Maaløes Vej 3, DK2200 Copenhagen N, Denmark. [15]Department of Life Sciences, SciLifeLab, Chalmers University of Technology, Gothenburg SE412 96, Sweden. ✉E-mail: hongzhonglu@sjtu.edu.cn; eduardk@chalmers.se

depends on cellular gene expression, which is determined by both the genotype and the environment. Diverse growth environments, encompassing variations in nutrition, temperature and stress, possess the capacity to strongly shape cellular metabolism, thereby exerting significant influence on the phenotypic outputs. Contrastingly, traditional GEMs are chiefly based on the whole genome sequence and their functional gene annotations (Thiele and Palsson, 2010). As a result, it is challenging for GEMs themselves to reflect gene expression levels corresponding to dynamic environmental changes. With the accumulation of various omics datasets, there has been a growing interest in the reconstruction of omics-constrained GEMs, especially in *Escherichia coli*, where the biological activity of metabolic networks is contextualized based on quantitative transcriptomics or proteomics (Angione and Lió, 2015; Becker and Palsson, 2008; Domenzain et al, 2022; Tian and Reed, 2018; Wagner et al, 2021; Martino et al, 2022). By contrast to ordinary GEM, context-specific GEMs are more powerful in simulating and revealing metabolic changes under environmental and genetic perturbations. However, until now, few studies have been conducted to systematically evaluate the quality and prediction capabilities of large-scale context-specific yeast-GEMs. On the other hand, model quality as denoted by e.g., accurate gene-protein-reaction relationships (GPR) and protein compartment annotations are at least as important in context-specific GEMs compared to ordinary GEMs (arguably even more, as incorrect annotations might become more influential when present in smaller subnetworks). The continuous improvement of yeast-GEM is therefore imperative for its use as both ordinary and context-specific GEMs.

To this aim, we released the latest version of the yeast-GEM (Yeast9) for the community by merging consistent model updates that have been made in the past five years. To display the value of yeast-GEM in transforming big data into knowledge, through leveraging the large-scale omics and phenotype datasets, we systematically reconstructed and analyzed numerous omics-constrained GEMs, i.e., 163 condition-specific GEMs (csGEMs) at single-cell level to decipher the metabolic readaptation mechanism under high osmotic stress and 1229 strain-specific GEMs (ssGEMs) to characterize the yeast metabolism under single gene deletion. These studies showcase that yeast-GEM is well-suited for conducting omics integrative analysis in order to uncover complex relations between genotype and phenotype. Moreover, yeast-GEMs can be used for predicting cellular responses to novel environmental conditions, as well as being computational platforms to pioneer the development of industrial strains. Therefore, Yeast9 can serve as a computational toolbox for quantifying yeast physiology and guiding experimental works for the wider yeast community.

# Results

## Model improvements from the yeast community

Through a collective effort of iterative engagements by the yeast-GEM community, the consensus yeast-GEM was updated from Yeast8 to Yeast9. Following a similar pipeline as employed for Yeast8, every round of updates was diligently recorded and comprehensively documented through a version-controlled system (https://GitHub.com/SysBioChalmers/yeast-GEM) which provided

transparency and reproducibility in the development of Yeast9. The coverage of the metabolic network was increased through a combination of targeted expansions, e.g., including reactions related to volatile esters & polyphosphates, and by identification of missing gene-protein-reaction relations (GPRs) from reaction databases KEGG (Kanehisa and Goto, 2000) and MetaCyc (Caspi et al, 2016). Combined with further curations described below, this yielded 29 new genes, 202 new reactions, and 139 new metabolites (Fig. 1).

Numerous GPRs and metabolite annotations were curated by reviewing the corresponding annotation from NCBI (Sayers et al, 2009), UniProt (The UniProt Consortium, 2017), KEGG, ChEBI (Hastings et al, 2016), PubChem (Kim et al, 2021), MetaNetX (Moretti et al, 2021), ModelSeed (Seaver et al, 2021), BiGG (King et al, 2016), BioCyc (Caspi et al, 2016) and Reactome (Jassal et al, 2020). We systematically curated all annotations of transport reactions according to the detailed protein function annotations at SGD (Cherry et al, 2012), TCDB (Saier et al, 2006), BioCyc, KEGG, and UniProt (Fig. 1), lending confidence to simulations involving transporter usage. The subunit composition of 36 protein complexes was corrected based on SGD, ComplexPortal (Meldal et al, 2022) and UniProt. To facilitate pathway analyses, each reaction was assigned to single explicit subsystems, according to the subsystem annotations (Dataset EV1) from KEGG, BioCyc, and the GO ontology in SGD. The top 20 subsystems are summarized in Fig. EV1.

When simulating flux distributions with the updated model, the feasibility of metabolic fluxes and their directionality can be determined by thermodynamics analysis. We therefore assigned $\Delta G^{o'}$ for 98.2% metabolites and 97.2% reactions according to evidence gathered from the yETFL model (Oftadeh et al, 2021), dGPredictor (Wang et al, 2021) and ModelSEED database. Furthermore, we balanced most mass/charge unbalanced reactions in the model, thus increasing the percentage of balanced reactions to 93.8%.

## Systematic evaluation of Yeast9 in its prediction quality

Through the above improvements, Yeast9 contains 2805 metabolites, 1162 genes, and 4130 reactions. Yeast9 has an improved performance in characterizing cell growth from a wide range of conditions compared to Yeast8. With a 27% increase in MEMOTE score (Lieven et al, 2020), the predictions of single gene essentiality and Biolog-plate measured substrate usage (Kang et al, 2019) by Yeast9 were moderately improved compared with Yeast8 (Fig. 2A,B). Predicted growth from Yeast9 correlated well with experimental data under both aerobic and anaerobic conditions with $R^2$ equals to 0.842 (Fig. 2C). Synthetic lethality can be predicted with almost 80% accuracy (Fig. 2D). Various false negative predictions may be the consequence of reactions that in Yeast9 are annotated with redundant isozymes, which may prevent in silico lethality. In vivo, however, the expression of isozymes may be transcriptionally regulated (Bradley et al, 2019; Zhang et al, 2018), which is an aspect that is not considered in GEMs. Worse synthetic lethality predictions were obtained when reactions were disabled even if the knockout gene was an isoenzyme, yielding an accuracy of less than 60% (Fig. 2E). This advocates for a more fine-grained consideration of isoenzyme activity, through e.g., the integration of condition-specific transcriptomics data, as

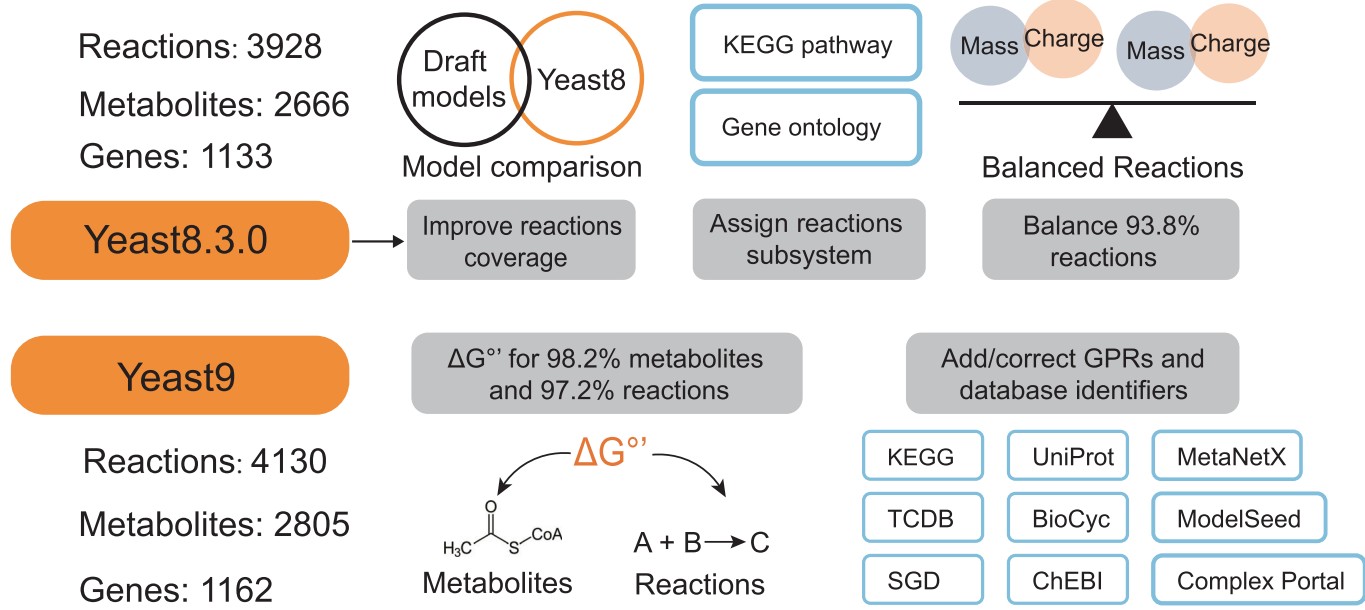

**Figure 1. Major improvements in Yeast9 compared to Yeast8.**

The Yeast9 model contains 1162 genes, 2805 metabolites, and 4130 reactions. New reactions were identified by comparing Yeast8 with draft models constructed by RAVEN. ΔG°' was added for almost all metabolites and reactions. Each reaction was linked with a single subsystem according to the pathway annotation from KEGG or SGD. Various GPRs were added or corrected by multi-rounds of manual comparison with databases. Nearly all reactions were curated to ensure mass and charge balances.

demonstrated below. It is also worth to mention that the low recall (13.1% in Fig. 2D, 37% in Fig. 2E) and precision (10.0% in Fig. 2D, 9.2% in Fig. 2E) highlight the limitations encountered by GEMs in predicting synthetic lethality.

The newly added thermodynamic information makes it possible to explore the driving force of mass transformation in metabolism at both single reaction and sub-pathway levels. The ΔG°' value is indicative of the thermodynamic driving force of a reaction under biochemical standard conditions, i.e., pH 7, 298 K, 1 atm and unit concentrations for chemicals other than water and protons. Nonetheless, as reaction directionality can be determined by calculating condition-specific in vivo ΔG values with $\Delta G = \Delta G°' + RT \ln Q$ (where Q is the ratio of product of reactant concentrations), the ΔG°' is still somewhat indicative of the likelihood of reaction reversibility.

In attempt to define reference ΔG values, we gathered average metabolite concentrations from the Yeast Metabolome Database (YMDB, Ramirez-Gaona et al, 2017), as this would allow the determination of Q. Only 125 of the model metabolites (or 315 when considering compartments) with reported concentrations can be found in YMDB. This low coverage precluded a systematic analysis of ΔG values, albeit the ΔG°' across metabolism can still be examined. In central carbon metabolism (Fig. 2F), pathways such as glycolysis, tricarboxylic acid cycle and pentose phosphate pathway have negative ΔG°' with −22.8, −9.6, and −13.8 kJ/mol, respectively. The ΔG°' of individual reactions, however, can range drastically, implying that metabolites reach concentrations (i.e., Q) that are compatible with a net flux through these reactions. This exemplifies the importance of considering metabolite concentrations before drawing conclusions on the effect of ΔG°' on the functioning of the metabolic network.

## Environmental adaptation mechanism revealed by single-cell omics-constrained GEMs

While yeast-GEM represents the theoretical metabolic network of *S. cerevisiae* based on its genome annotation, not all enzymes might be constitutively expressed. Gene expression may differ between cells, and consequentially the metabolic network of individual cells may not be the same. To examine this, we collected 163 single-cell transcriptomes of *S. cerevisiae* (Gasch et al, 2017; Data ref: Gasch et al, 2017), including 80 transcriptomes measured under high osmotic stress and 83 transcriptomes measured under reference conditions. We used GIMME (Becker and Palsson, 2008) to construct 163 single-cell omics-constrained GEMs (scGEMs) by modifying the presence of reactions and metabolites in the model based on transcriptomic data (Fig. 3A). Due to the nature of single-cell data it cannot be distinguished whether the scGEMs reflect true inter-cell variability, or rather only reflect stochasticity in the single-cell data acquisition, a challenge encountered in any single-cell approach. The generated scGEMs have reaction numbers ranging from 2223 to 3856, with the numbers of metabolites ranging from 2088 to 2708.

We gathered the metabolite number, reaction number, and projected flux of each scGEM to categorize *S. cerevisiae* cells sampled from stressed and unstressed conditions (Fig. 3A). Kernel principal component analysis (kPCA) was executed to extract features before machine learning. A random forest classification model was trained using the kPCA-processed data. After parameter optimization, the random forest classifier demonstrated 78% accuracy in differentiating single cell sampled between the osmotic stress and unstressed conditions (Fig. 3B). The optimized parameters are presented in Dataset EV2. The same analysis on the whole transcriptomics dataset yielded 100% accuracy (Fig. EV2), corroborating the observations by

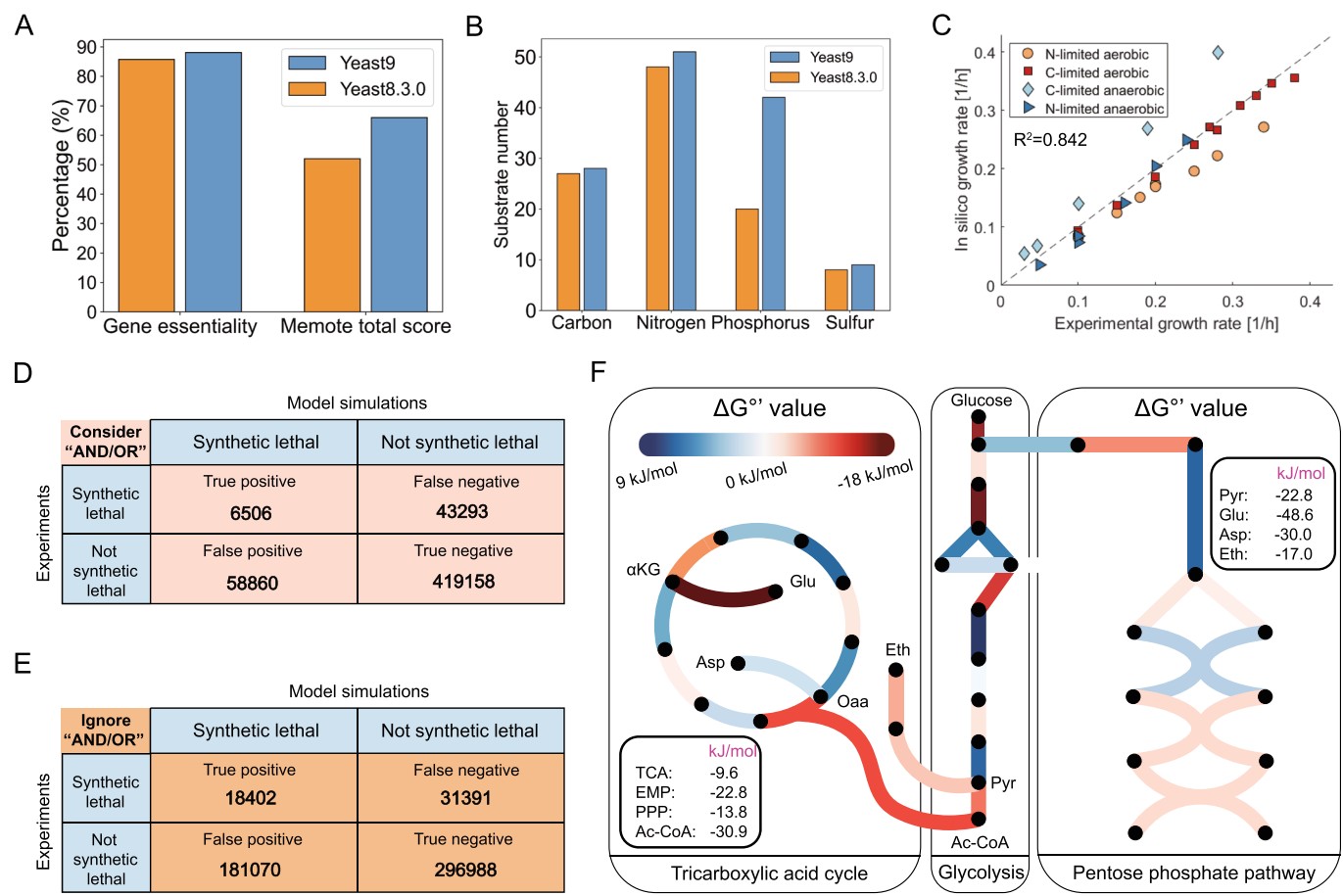

**Figure 2. Systematic evaluation of prediction capability by Yeast9.**

(A) Comparison in predicted gene essentiality and Memote score between Yeast8 and Yeast9. (B) Carbon, nitrogen, phosphorus and sulfur source usages comparison between Yeast8 and Yeast9. (C) Growth simulation under aerobic and anaerobic conditions. (D) The Yeast9 could predict the consequences of synthetic lethality of two gene combinations considering the "AND/OR" relationship, with the accuracy at 80%. (E) Ignoring the "AND/OR" relationship, the Yeast9 shows 59% accuracy in predicting synthetic lethality. (F) The profile of ΔG°' in TCA, EMP, and PPP. The color denotes ΔG°' value. The red line means that the reactions are favorable thermodynamically; the blue line indicates that the reactions are unfavorable thermodynamically. The number within the bold rounded rectangle represents the total ΔG°' of TCA, EMP, PPP, and reaction pathways in synthesizing acetyl-CoA, pyruvate, glutamine, aspartate, and ethanol from glucose.

Gasch et al (2017). This implied that the metabolic networks of individual cells were sufficiently different to enable categorizing them by which environment they resided without considering the differential expression of non-metabolic genes. However, differential expression of non-metabolic genes are important determinants of the osmotic stress state that likely have strong biological relevance.

Cluster analysis on the simulated fluxes further illustrated high similarity within the same condition, while retaining heterogeneity (Fig. 3C). To further analyze the mechanism by which *S. cerevisiae* responds to osmotic stress conditions, single-cell specific growth rates (Fig. 3D) and central carbon metabolism fluxes (Fig. 3E; Dataset EV3) were calculated by simulating the scGEMs. While *S. cerevisiae* generally grew slower under osmotic stress, a subset of stressed cells grew similar to those in unstressed cells and vice versa. In this scenario, the alterations in the scGEMs' metabolic network topologies, driven by single-cell transcriptomes, result in growth heterogeneity that accurately reflects the true cellular growth state. This shows a possible existence of a resistant phenotype or alternative stress response pathways that might be

worth exploring. In terms of evolution, it is possible that heterogeneity of fluxes at the single-cell level play a role in aiding *S. cerevisiae* populations in their adaptation to new environments.

In a more detailed analysis, we examined the flux distributions from stressed and unstressed cells, particularly focusing on the tricarboxylic acid cycle (TCA), pentose phosphate pathway (PPP), and glycolysis (Fig. 3E). About 56% of the active fluxes are divergent between unstressed cells and stressed cells (Fig. EV3), in central carbon metabolism all reactions were significantly distinct. An increased flux through PPP, TCA, and lower-glycolytic fluxes was found in stressed yeast cells, which is consistent with proteomics data (Soufi et al, 2009), which show that the overall protein expression of these three pathways are changed in the same direction as the flux. This signifies that *S. cerevisiae* strengthens its central carbon metabolism to generate more ATP, provide NADPH redox potential, and possibly synthesize more precursors in response to high osmotic stress. Furthermore, the stressed scGEMs had an increased flux from dihydroxyacetone phosphate (DHAP) via glycerol 3-phosphate to glycerol, an osmoprotectant whose

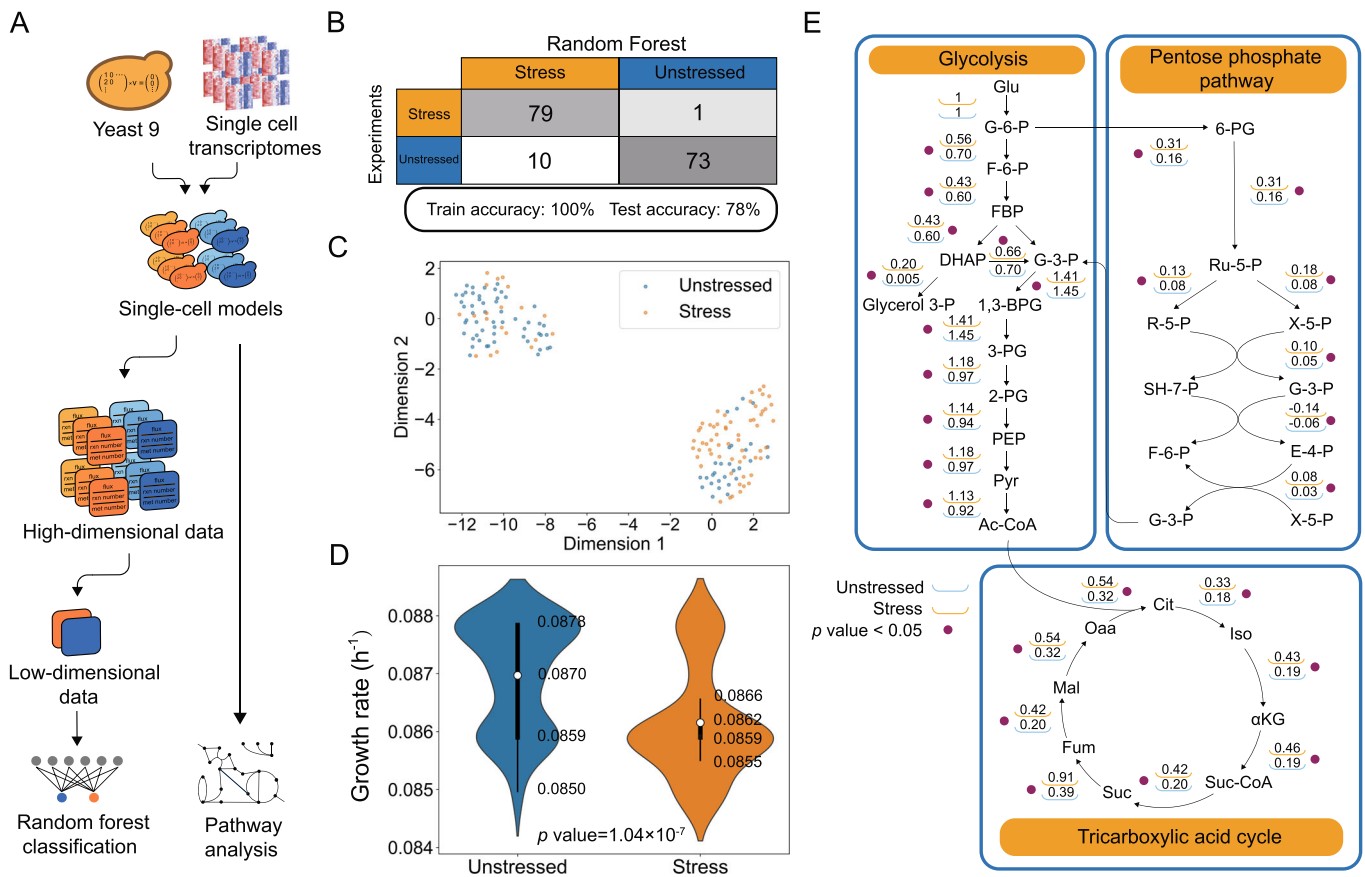

**Figure 3. Construction and application of scGEMs derived from Yeast9.**

(A) Construction and analytic workflows. We integrated single-cell transcriptomics measured under osmotic stress conditions and normal conditions into yeast-GEM by GIMME, resulting in 163 scGEMs. (B) The random forest classifier has 78% accuracy in classifying the single cell from high osmotic stress and normal conditions. (C) UMAP is used for cluster analysis, with dimension 1 plotted against dimension 2. (D) Predicted growth rates of single cells from osmotic stress conditions and normal conditions. $p$ value was analyzed by Student's $t$ test; white circle indicates mean; thick bar indicates interquartile range; thin bar indicates 1.5× interquartile range; $n = 80$ for stress and 83 for unstressed. (E) Comparison of mean flux values for central carbon metabolism as predicted by scGEMs. $P$ values were analyzed by Student's $t$ test. See Dataset EV3 for a full list of $P$ values. $n = 80$ for stress and 83 for unstressed.

biosynthesis during osmostress is well documented (Hohmann, 2009; Yale and Bohnert, 2001).

## Multi-omics analysis elucidates the metabolic rewiring under nitrogen limitation

Another environmental parameter affecting metabolism is nutrient availability, and as has been reported, *S. cerevisiae* preferentially assimilates ammonium or certain amino acids, specifically glutamine and glutamate (Crépin et al, 2012; Hofman-Bang, 1999). To check whether yeast-GEM could quantitatively classify the preferred nitrogen source utilized by *S. cerevisiae*, preference scores for the nitrogen sources ammonium, glutamate, isoleucine and phenylalanine were calculated using Yeast9 (Fig. 4A). During long-term evolutionary processes, the cellular resources required for nitrogen uptake and utilization influences *S. cerevisiae* preference for nitrogen sources. *S. cerevisiae* evolutionarily chooses a nitrogen source that is more resource-efficient and easier to utilize. If only limited nitrogen sources are available, *S. cerevisiae* could contrastingly evolve more efficient use of those nitrogen sources. Irrespective, as glucose acts as the main

carbon skeleton and energy source for yeast, we defined the nitrogen preference score is the absolute value of the slope between nitrogen and glucose uptake rates, where the uptake of a non-preferred nitrogen source will result in a more significant increase in glucose uptake when compared to a preferred nitrogen source. The order of preferred nitrogen sources (Fig. 4B) based on Yeast9 predictions were consistent with previous studies (Yu et al, 2021b; Hofman-Bang, 1999).

To further investigate how nitrogen sources tune yeast metabolism, integrative analysis with Yeast9 was carried out by leveraging reported multilayer omics datasets (Yu et al, 2021a, 2020). As the first step, we calculated flux distributions by constraining Yeast9 with measured exchange fluxes and total protein concentrations. As shown in Fig. 4C, nitrogen sources largely reshape cellular flux distribution. Next, we analyzed consistent tendencies (i.e., not correlations, but whether directions of change agreed) between reaction fluxes and the related protein abundances (Data ref: Yu et al, 2021a), comparing the unpreferred nitrogen sources (isoleucine or phenylalanine) with the favored one (i.e., ammonia). In these two cross-comparisons, 164 (isoleucine) and 166 (phenylalanine) of the Yeast9 genes show consistent

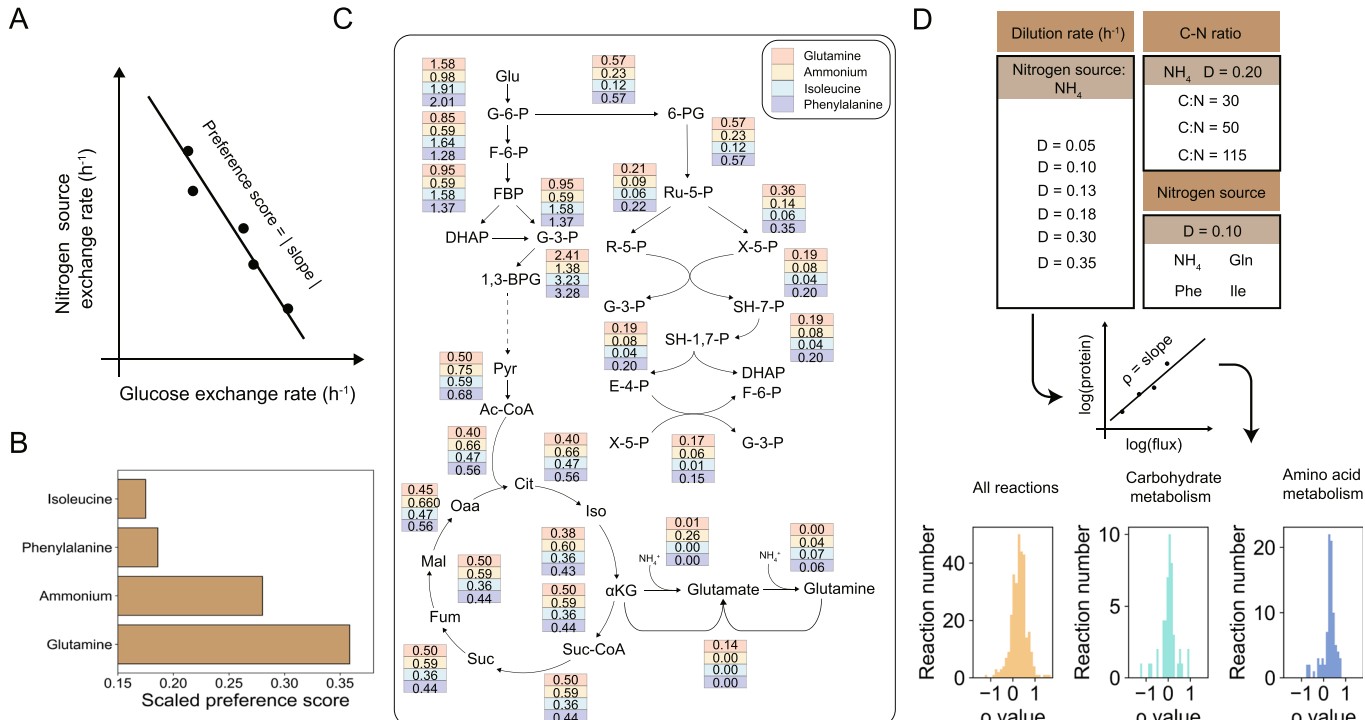

**Figure 4. Illustration of yeast metabolism rewiring under nitrogen limitation based on Yeast9 and multi-omic integrative analysis.**

(A) A graphical representation of the preference score. The preference score denotes the degree to which the change in glucose uptake rate compensates for the decrease in nitrogen source absorption rate. (B) The preference scores of glutamine, ammonium, isoleucine, and phenylalanine. (C) pFBA was used to get the flux distributions of four nitrogen sources. (D) 12 phenotype-constrained models and the calculation of protein regulation coefficient (ρ). The correlation analysis between protein abundances and fluxes was conducted at both holistic and sub-pathway levels.

directional change in both flux and protein level (Table EV1). There are 149 common genes selected from the above two groups of genes, which were enriched in amino acid biosynthesis (*P* value < 0.001) by GO term enrichment analysis (Dataset EV4).

To more precisely determine the covariance between fluxomics and proteomics across diverse environments, we expanded the analysis to not only consider the direction of change in both flux and protein level, but to evaluate if protein and flux levels quantitatively correlated across multiple conditions. We thereby utilized 12 phenotype-constrained models (Fig. 4D) and their associated proteomic datasets (Data ref: Yu et al, 2021a, 2020), covering not only alternative nitrogen sources but also six dilution rates and three carbon-nitrogen ratios, in a regulatory analysis that has previously been described for *E. coli* (Kochanowski et al, 2021). The contribution of protein expression to flux can be calculated as the slope between log (flux value) and log(protein abundance) (Fig. 4D), which has been defined as the protein regulation coefficient (ρ) (Rossell et al, 2006; Chubukov et al, 2013). Here, ρ ≈ 1 signifies that changes in simulated fluxes can largely be explained by protein concentration changes. When based on fluxes predicted from parsimonious FBA (pFBA), most reactions from carbohydrate metabolism and amino acid metabolism exhibited weak protein regulation coefficients (Fig. 4D), with only 1.7% reactions revealing high coefficients (ρ > 0.5, Table EV2). As pFBA only yields single flux distributions, ρ values were also determined based on mean fluxes from random sampling of the solution space, which yielded similar results (Table EV2). Both analyses imply a low correlation between protein abundance and flux, albeit it cannot be excluded that

uncertainty of the predicted fluxes obscure the true correlation between protein abundance and flux. Regardless, the previous analysis implies that the direction of change might be more distinctly conserved. Additional post-translational regulatory mechanisms might be implicated in this discrepancy. As proteome changes are not necessarily reflected in flux changes, multi-omics analyses are rendered more valuable to reflect metabolic adaptation mechanisms.

## Large-scale transcriptomic empowers yeast-GEM to predict growth profiles and gene function

When not lethal, gene knockouts may still alter gene expression levels, flux distributions and thereby phenotypes of an organism. Meanwhile, the knockout of genes that have similar functions may also yield similar changes in flux distributions. Therefore, it is possible to explore gene functions by analyzing model-predicted fluxes and gene expression levels upon gene knockouts. To this purpose, we collected two transcriptomic datasets containing 1143 single knockout *S. cerevisiae* strains (Kemmeren et al, 2014; Sameith et al, 2015; Data ref: Kemmeren et al, 2014; Data ref: Sameith et al, 2015) and 86 single or double knockout strains to build 1229 ssGEMs using the early described method (Culley et al, 2020), where reaction bounds are changed according to gene expression levels (Fig. 5A). Of these 1143 knockout genes, 75 were assigned to reactions in Yeast9, implying that the ssGEMs are mostly reflecting the metabolic networks after knockout of non-enzyme-coding genes. Flux balance analysis (FBA) with growth

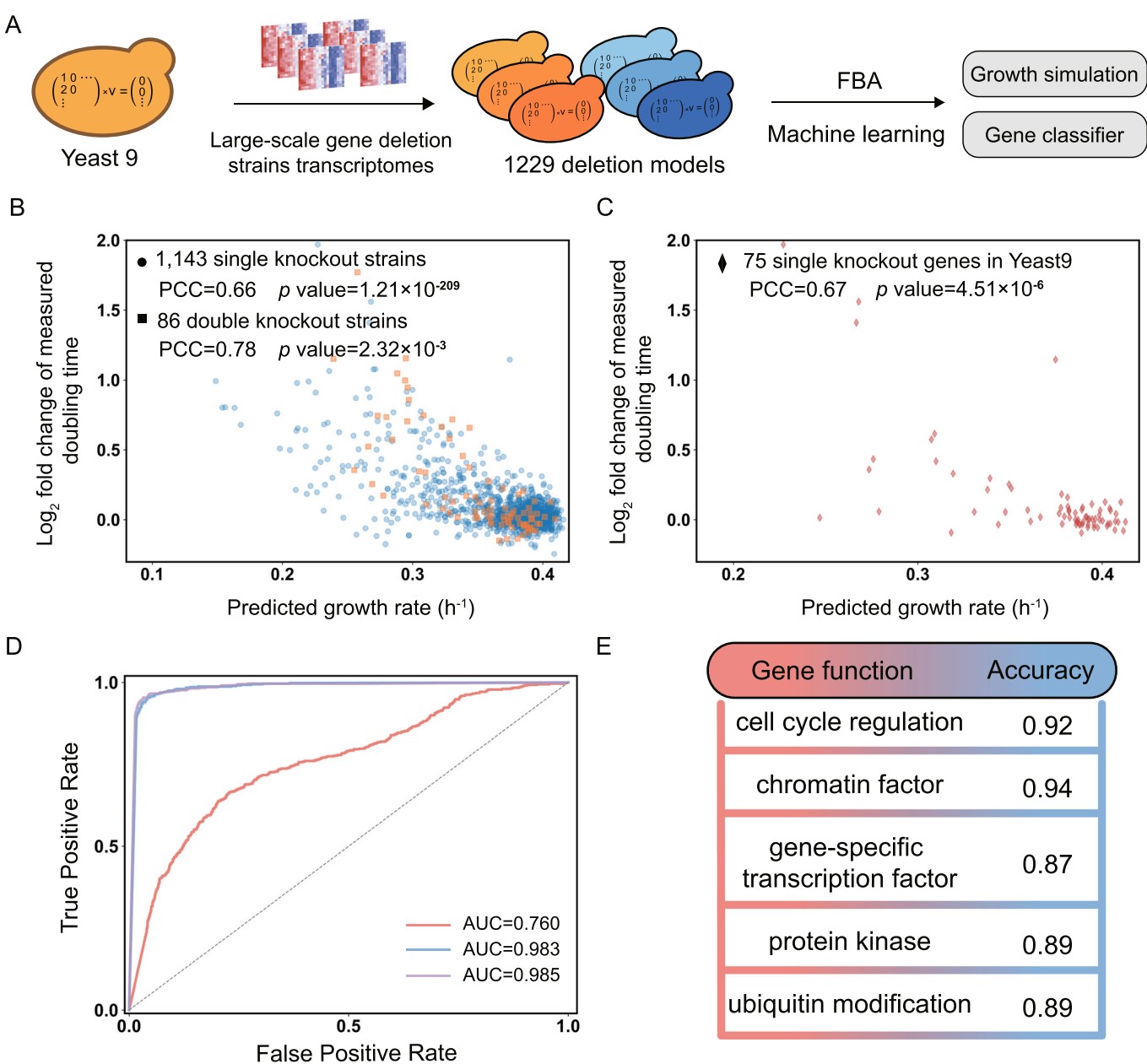

**Figure 5. Large-scale transcriptomics-constrained GEMs built from Yeast9 could characterize the growth profiles of gene knockout strains and classify gene functions.**

(A) The Yeast9 is constrained by transcriptomics to generate ssGEMs for gene knockout strains, which are sequentially used to predict the growth rate and train machine learning models in gene function classification. (B) Significant correlations existed between the relative doubling time (calculated based on measured value) and the corresponding simulated growth rate based on ssGEMs. $P$ values were analyzed by Student's $t$ test. Correlation was analyzed by Pearson correlation coefficient. (C) Among 1143 single knockout genes, 75 genes present in Yeast9. A distinct correlation is evident between the relative doubling time and the simulated growth rate. $p$ values were analyzed by Student's $t$ test. Correlation was analyzed by Pearson correlation coefficient. (D) The ROC and AUC of Naive Bayes using fluxomic (blue line), transcriptomic data (orange line) and the combination data (purple line) were computed by scikit-learn. (E) Using the Naive Bayes classifier, the predictive accuracy for each gene function in the test set was evaluated.

maximization as the objective function showed a good correlation between the predicted and measured growth rate, with Pearson correlation coefficient (PCC) = 0.66 for single knockout strains and 0.78 for double knockout strains (Fig. 5B). The PCC reached 0.67 when only the 75 genes that appear in Yeast9 are considered (Fig. 5C). In comparison to Culley et al (2020), which used the pre-

Yeast8 *S. cerevisiae* GEM iSce926 that had explicitly been curated for synthetic lethality analysis (Chowdhury et al, 2015), Yeast9 exhibited enhanced PCC values for strains with double knockouts. Nonetheless, the relatively moderate improvement in predictive performance is representative of the gradual improvement of yeast-GEM through each subsequent release. Meanwhile, it should

be noted that the expression of metabolic genes (and their corresponding constraints on the metabolic model) is not the sole factor that dictate strain-specific growth rates, as growth rate predictions based solely on the whole transcriptomics dataset reached PCC > 0.9 (Culley et al, 2020).

To evaluate the ability to annotate gene function based on flux simulations, the aforementioned 1143 knocked-out genes were assigned to functional categories (one per gene), in accordance with Culley et al (Dataset EV5). In order to provide sufficient training data, we selected the five functional categories with the highest number of genes: cell cycle regulation; chromatin factors; gene-specific transcription factors; protein kinases; and ubiquitin(-like) modifications, together covering 559 genes (Table EV3), including 11 genes assigned to reactions in yeast-GEM, and assigned these classifications to the aforementioned knockout ssGEMs. We then explored the application of machine learning algorithms for this five-class classification, utilizing diverse datasets, including transcriptome data, fluxomic, and their concatenated composites that were preprocessed by kPCA.

Five machine learning algorithms were tested, including Support Vector Machine (SVM), Naive Bayes, Random Forest, K-Nearest Neighbors (k-NN), Logistic Regression, and Multilayer Perceptron (MLP), while a systematic optimization of parameters was conducted for each model (Dataset EV2). The Naive Bayes classifiers showed the best performance with 90% testing accuracy when utilizing both fluxomics and transcriptomics as input data (Table 1). The testing performance (Table 1), as well as the receiver operating characteristic curve (ROC) and the area under the curve (AUC) all show that the Naive Bayes classifiers performed better when utilizing fluxomic as the only input compared to transcriptomics (Fig. 5D). In the test set, the prediction accuracy of chromatin factor is the highest, followed by cell cycle regulation (Fig. 5E). In addition, excluding 11 genes that are included in yeast-GEM from the analysis did not drastically change the performance (Table EV4). Collectively, it suggests that fluxes from the ssGEMs can provide sufficient information to classify the function of knocked-out genes, thus helping to bridge the gap between genotype and phenotype.

While demonstrating the feasibility of this approach and the suitability of yeast-GEM, this framework could gain impact with more knockout transcriptomics, single-function gene annotations, or machine learning approaches that allow for multi-label classification, as would be required to handle, e.g., GO term annotations.

## Discussion

Through a concerted effort by members of the research community, we have updated yeast-GEM from Yeast8 to Yeast9, assisted by a version control system. The gradual improvements of yeast-GEMs covered a wide range of curations, including but not limited to the assignment of new gene functions; adjustment of incorrect gene assignments; modification of reaction directionality; and inclusion of annotations of proteins, metabolites and reactions. The aforementioned progress has filled gaps in the current yeast-GEM and enhanced its performance in comprehensively characterizing yeast cellular metabolism. However, some limitations still exist for Yeast9. For example, it still lacks high-resolution details in representing yeast metabolic activities at the organelle level and some reactions for lipid synthesis are not standard. Our synthetic

**Table 1. Classifier metrics when trained on different datasets, aimed at achieving a five-class gene function classification.**

| Dataset | Algorithm | Train data accuracy | Test data accuracy | Test data recall rate | Test data F score |
|---|---|---|---|---|---|
| Transcriptomic | SVM | 0.99 | 0.68 | 0.66 | 0.67 |
| | MLP | 1.00 | 0.67 | 0.66 | 0.66 |
| | Naive Bayes | 0.58 | 0.32 | 0.35 | 0.30 |
| | Random forest | 1.00 | 0.57 | 0.57 | 0.56 |
| | k-NN | 1.00 | 0.60 | 0.58 | 0.60 |
| | Logistic | 0.99 | 0.72 | 0.70 | 0.70 |
| Flux | SVM | 0.52 | 0.39 | 0.35 | 0.33 |
| | MLP | 0.34 | 0.31 | 0.27 | 0.17 |
| | Naive Bayes | 0.92 | 0.88 | 0.89 | 0.88 |
| | Random forest | 0.99 | 0.70 | 0.69 | 0.70 |
| | k-NN | 1.00 | 0.57 | 0.54 | 0.55 |
| | Logistic | 0.47 | 0.32 | 0.28 | 0.24 |
| Transcriptomic and Flux | SVM | 0.84 | 0.53 | 0.51 | 0.51 |
| | MLP | 0.28 | 0.30 | 0.30 | 0.28 |
| | Naive Bayes | 0.94 | 0.90 | 0.90 | 0.90 |
| | Random forest | 1.00 | 0.83 | 0.82 | 0.83 |
| | k-NN | 1.00 | 0.43 | 0.41 | 0.41 |
| | Logistic | 0.55 | 0.37 | 0.36 | 0.36 |

lethality analysis yielded almost 80% accuracy of prediction, leaving room for improvement. Thus, further efforts from the yeast community still need to be taken as part of the circular process to refine the quality of Yeast9.

From a mathematical perspective, the computational output of GEMs represents substantial solution spaces containing large numbers of potential flux distributions. While mathematically sound, not every flux distribution is as likely to be biologically observed, given that biological variability simultaneously deviates from the principle of FBA. As conventional GEMs neglect the effects of mRNA and protein levels on the shape of the solution space, discrepancies between model simulations and real metabolic activities are a consequence. We therefore evaluated Yeast9 in a number of integrative multi-omics analyses that quantified yeast physiology and metabolism. As the first example, 163 scGEMs could be classified according to their exposure to osmotic stress and showed subpopulations when considering their metabolic networks. Secondly, Yeast9 was able to enumerate which nitrogen sources (i.e., glutamine and ammonium) are most preferred by yeast. In additional simulations with Yeast9, different nitrogen sources could drastically alter the fluxes through central carbon metabolism. When considering integrative multilayer omics analyses with Yeast9, we observed a low consistency between changes in fluxomics and proteomics. Such a lower correlation has also been reported in other microorganisms, e.g., *E. coli* and *Bacillus subtilis* (Gerosa et al, 2015; Hackett et al, 2016; O'Brien et al, 2016; Chubukov

et al, 2013). Thereby, multi-omics analyses and GEM simulations are highly complementary when investigating metabolic regulation. Omics-constrained GEMs reduce the solution space, thereby eliminating biologically infeasible solutions and consequentially resulting in model outcomes with higher confidence. At the same time, this does not mean that it is strictly essential to integrate omics data in Yeast9 before simulations can be performed, just as with any other genome-scale model of metabolism. The ML approaches demonstrated in this manuscript obligate transcriptomics data, thereby highlighting a potential limitation of these approaches, as such data is not always readily available.

Overall, Yeast9 is the most comprehensive and state-of-the-art *S. cerevisiae* GEM, as well as a valuable knowledge database on its metabolic network. Through continuing iterative updates, the quality of yeast-GEMs has further improved and as such, this model could function as valuable template when generating high-quality GEMs for non-conventional yeast species, such as *Pichia pastoris*, *Ogataea polymorpha* and Methylotrophic yeasts (Domenzain et al, 2021; Grigaitis et al, 2022; Liebal et al, 2021; Tomàs-Gamisans et al, 2018). Those models together can enable the exploration of evolutionary mechanisms underlying diverse metabolic activities and traits across yeast species. Ultimately, we are confident that the latest version of yeast-GEMs–Yeast9 and its flourishing model ecosystem, which still needs research community to contribute to further improvements, fostering an open-source and collaborative environment, around it will accelerate the developments in systems and synthetic biology studies of yeast in the coming years.

# Methods

### Reagents and tools table

| Reagent/ resource | Reference or source | Identifier or catalog number |
|---|---|---|
| **Software** | | |
| yeast-GEM 9.0.0 | https://github.com/SysBioChalmers/ yeast-GEM | |
| MATLAB R2021b | https://www.mathworks.com/ products/matlab.html | |
| RAVEN 2.9.2 | https://github.com/SysBioChalmers/ RAVEN | |
| COBRA 3.0.1 | https://github.com/opencobra/ cobratoolbox | |
| python 3.9.7 | https://www.python.org | |
| cobrapy 0.22.1 | https://github.com/opencobra/ cobrapy | |
| umap-learn 0.5.3 | https://github.com/lmcinnes/umap | |
| scikit-learn 0.24.2 | https://scikit-learn.org | |
| memote 0.13.0 | https://github.com/opencobra/ memote | |
| scipy 1.7.1 | https://scipy.org | |
| numpy 1.20.3 | https://numpy.org/ | |
| pandas 1.3.4 | https://pandas.pydata.org | |

## Model curation by identifying new reactions, metabolites, and genes

Standard procedures for metabolites and reactions annotation used in this work were consistent with Yeast8 (Lu et al, 2019).

- Aimed at adding new reactions and metabolites, two draft models were reconstructed using RAVEN Toolbox 2.0 (Wang et al, 2018). The two draft models were built based on KEGG and MetaCyc separately (Kanehisa and Goto, 2000; Caspi et al, 2016).
- Then, new reactions and metabolites were extracted by semi-automatically comparing the Yeast8 with those two draft models.
- The detailed annotation of genes, metabolites and reactions from MetaCyc, Yeastcyc, UniProt, SGD, and KEGG were utilized to guarantee that the new reactions and metabolites were reasonable and of high quality.

## Model curation by adding reaction subsystems

Reaction subsystems were systematically obtained from the KEGG pathway and integrated into the Yeast9.

- If KEGG does not have the subsystem annotation of the reactions, the Gene Ontology of the corresponding genes in SGD or basic biochemistry knowledge was applied.
- If no detailed information is found, the reactions are classified into the "other" subsystem.

## Model curation by adding Gibbs free energy

The Gibbs free energy change ($\Delta G°$) was added into yeast-GEM. The $\Delta G°$ values in kJ/mol are available in the YAML version of the model, and when loaded through the provided *loadYeastModel* function, the values are available from the metDeltaG and rxnDeltaG fields.

- Whenever possible, the $\Delta G°$ values for reactions (97%) and metabolites (96%) were gathered from the yETFL model (Oftadeh et al, 2021).
- The $\Delta G°$ for reactions that were not contained in the yETFL model were computed by dGPredictor (Wang et al, 2021).
- The $\Delta G°$ for metabolites that were not contained in the yETFL model were taken from ModelSEED (Seaver et al, 2021).

## Curation of complex annotations

The complex annotations were comprehensively refined, mainly based on ComplexPortal, and information from SGD and UniProt were also used.

## Curation of transporter annotations

The transporter annotations were curated using various databases, including TCDB, SGD, UniProt, and KEGG.

## Synthetic lethal simulation

Gene interactions data (Costanzo et al, 2016; Tong et al, 2004; Bellaoui et al, 2003; Goehring et al, 2003; Huang et al, 2002; Kozminski et al, 2003; Krogan et al, 2003; Tong et al, 2001) were collected.

- The function "double_gene_deletion" in COBRA Toolbox was used to estimate synthetic lethality considering the "AND/OR" relationship.
- For the estimation ignoring "AND/OR" relationship, all reactions of a deleted gene were removed no matter if other genes are associated with those reactions and how.

## scGEMs construction and simulation

Gasch et al previously described a series of single-cell transcriptomes of *S. cerevisiae*, in which 80 transcriptomes were measured under the osmotic stress condition and 83 transcriptomes under the unstressed condition (Gasch et al, 2017).

- Constrained by the single-cell Gasch's transcriptome dataset, the 163 single-cell-specific models were generated using the GIMME algorithm by COBRA Toolbox (Becker and Palsson, 2008) in MATLAB based on Yeast9.
- To analyze the metabolic difference between the osmotic stress condition and the unstressed condition, parsimonious flux balance analysis (pFBA) (Lewis et al, 2010) was used to maximize growth rate under the minimal medium condition.
- The resulting 163 flux distributions, together with the reaction numbers and metabolite numbers, formed 163 datasets.
- Subsequently, the dataset was randomly split into training (70%) and test (30%) datasets. The parameters of kPCA ('n_components' and 'kernel') as well as those of the Random Forest classifier ('n_estimators', 'max_depth', 'min_samples_split', and 'min_samples_leaf') were optimized using the GridSearchCV function from the scikit-learn library (Pedregosa et al, 2011) along with a fivefold cross-validation
- UMAP was utilized to reduce the dimensions of 163 datasets and to perform cluster analysis, where the n_components=2 (McInnes et al, 2020).

## Generation of condition-specific GEMs under nitrogen limitation

- Yeast9 was constrained by the experimentally measured growth rate and exchange rates (except for nitrogen exchange rate) under nitrogen limitation conditions (Yu et al, 2021b, 2020).
- The carbohydrate, protein, and RNA ratios in the biomass composition were scaled according to the measured carbohydrate, protein, and RNA abundance in the paper. As a result, 12 phenotype-constrained models were generated.

## Compute nitrogen source preference score

The preference score of yeast for different nitrogen sources was computed as follows:

- Step 1: Allow for uptake of one nitrogen source at the time, with a fixed growth rate of $0.1\,h^{-1}$ (i.e., the experimental growth rate).
- Step 2: Determine the minimum nitrogen uptake by FBA while setting the relevant nitrogen source exchange reaction as the objective function.
- Step 3: Step-wise (in five steps) increase the nitrogen uptake from 100 to 150% of the value determined in step 2, and determine the minimum glucose uptake by FBA while setting the glucose

exchange reaction as objective function.
- Step 4: Determine the slope of glucose uptake versus nitrogen source uptake, and take its absolute value to represent the preference score.
- Step 5: The scores are scaled by Eq. (1).

$$\text{scaled}\,i = \frac{i}{\sum I} \qquad (1)$$

Where $i$ denotes the origin preference score. $I$ denotes a vector containing all scores.

## Compute flux distribution under nitrogen limitation

- To get the flux distribution under four nitrogen sources, Yeast9 was first constrained by the measured fluxes (multiplied by 0.8 considering the possible experimental error) for each related exchange reaction and the measured growth rate, with minimization of nitrogen source utilization as the objective function.
- Subsequently, the uptake rate of nitrogen source in models was fixed, to smaller than 1.5 times of the calculated minimal nitrogen source uptake rate.
- Afterwards, all fluxes were recalculated by pFBA with maximizing growth as the objective function.

## GO enrichment analysis under nitrogen limitation

- The genes with higher protein expression level and corresponding flux than their counterpart in the model using $NH_4$ as the only nitrogen source were selected. For genes related to more than one reaction, at least one related reaction flux larger than that in the NH4 model meets screening requirements.
- After that, GO enrichment analysis (https://david.ncifcrf.gov/) was used to estimate the biological processes in which the selected genes were involved.

## Protein regulation coefficients from fluxomics and absolute proteomics

Proteomic data and yeast phenotype measurements were collated from literature (Yu et al, 2021a, 2020; Data ref: Yu et al, 2021a, 2020).

- Yeast9 was constrained by the measured exchange fluxes and growth rates. The biomass composition was altered according to total protein concentrations from the aforementioned publications, using the "scaleBioMass" function in the yeast-GEM repository. Then, nitrogen uptake (for which no measurements were available) was minimized and subsequentially fixed to the obtained value.
- Lastly, the flux distribution was calculated by pFBA (Kochanowski et al, 2021) with maximized ATP maintenance as objective (results used in Fig. 4D and Table EV2) or by taking the mean value of 1000 random samples of the solution space (Bordel et al, 2010) (results used in Table EV2).
- To determine the protein regulation coefficient ρ, only reactions and proteins that showed nonzero value in a minimum of five out of twelve conditions were considered. The ρ value was determined for each reaction-protein pair individually for each limitation by linear regression between the log-fluxes and log-protein concentrations. When multiple isoenzymes were associated with a reaction, the average regulation coefficient of all corresponding proteins was computed to determine the final protein regulation

coefficient for that reaction. For reactions catalyzed by a complex, the minimal concentration of all subunits was used to compute the protein regulation coefficients.

## Defining strain-specific GEMs based on single/double knockout transcriptomics

The transcriptomic data of single- and double-knockout strains (Kemmeren et al, 2014; O'Duibhir et al, 2014; Sameith et al, 2015) were used to constrain reaction lower and upper bounds, by utilizing the algorithm and MATLAB code previously described by (Culley et al, 2020).

- Briefly, reaction lower and upper bounds as defined in Yeast9 were defined as representing the wildtype reference strain.
- Then, non-log-transformed gene expression levels as obtained from microarray experiments (ranging from 117-fold down-regulation to 64-fold upregulation in comparison to wildtype) were used to multiply the existing lower and upper bounds of gene-associated reactions.

To prevent unrealistic flux bounds, winsorization was applied to smooth extreme values (except for the knockout gene), to make them fit within the 1st and 99th percentile of gene expression values. If a reaction was annotated with multiple genes, the minimum expression level among subunits, and/or the maximum expression level among isoenzymes were used. Through this approach, the original solution space was reshaped to represent the knockout strain-specific solution spaces.

## Gene function prediction using 6 machine learning algorithms

The deleted genes were classified according to the PANTHER classification system (Mi et al, 2019) in a previous study (Culley et al, 2020). In this dataset, the gene number in those functional categories ranges from 29 to 149.

- The categories with the top five highest number of genes (cell cycle regulation, chromatin factors, gene-specific transcription factors, protein kinases, and ubiquitin(-like) modifications) were selected for machine learning classification.
- A suite of machine learning models, including SVM, MLP, Naive Bayes classifiers, Random Forests, k-NN, and Logistic Regression, was applied to categorize the genes.
- The dataset involved the predicted fluxes and transcript profiles, along with the combination, segmented stochastically into training (70%) and testing (30%) subsets.
- kPCA was used for extracting omics features.
- The optimization of the model's hyperparameters was executed through the GridSearchCV method, with a fivefold cross-validation routine. The assessment of these models was grounded on several performance metrics including accuracy, recall, F1 score, ROC curve and AUC.

## Data availability

The datasets and computer code generated in this study can be accessed through the following databases: Yeast9 computer scripts:

GitHub (https://GitHub.com/SysBioChalmers/yeast-GEM). Model evaluation and omics integrative analysis: GitHub (https://GitHub.com/hongzhonglu/yeast_GEM_multi_omics_analysis). Supplementary files to construct knockout models: Figshare (https://figshare.com/articles/dataset/large_data_used_in_https_github_com_hongzhonglu_yeast_GEM_multi_omics_analysis_/22774076).

The source data of this paper are collected in the following database record: biostudies:S-SCDT-10_1038-S44320-024-00060-7.

## Peer review information

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

## Acknowledgements

This work is supported by grant 2022YFA0913000 from the National Key R&D Program of China, Shanghai Pujiang Program, and grants 22208211 and 22378263 from the National Natural Science Foundation of China (NSFC). This work is also supported by the Novo Nordisk Foundation (grant no. NNF20CC0035580); the Knut and Alice Wallenberg Foundation, and the European Union's Horizon 2020 research and innovation program (grant agreements 686070 and 720824); National Key Research and Development Program of China (2020YFA0907800); the 111 Project (B18022); Dutch Research Council (Nederlandse Organisatie voor Wetenschappelijk Onderzoek (NWO)) for the UNLOCK initiative (NWO: 184.035.007); CN Yang Scholars Programme; Deutsche Forschungsgemein-schaft (DFG, German Research Foundation) under Germany's Excellence Strategy—Cluster of Excellence 2186; Centro de Modelamiento Matemático, ACE210010 and FB210005, BASAL funds for Centers of Excellence from ANID-Chile Project ICN2021 044 of the Millennium Scientific Initiative Grant Exploración number 13220002; and CONICYT Becas Chile grant #72180373 (https://www.conicyt.cl/becasconicyt/). The funding bodies had no role in the design of the study, analysis and interpretation of the data, preparation of the manuscript, and decision to submit the manuscript for publication.

## Author contributions

**Chengyu Zhang**: Formal analysis; Investigation; Writing—original draft; Model curation. **Benjamín J Sánchez**: Writing—review and editing; Model curation. **Feiran Li**: Writing—review and editing; Model curation. **Cheng Wei Quan Eiden**: Writing—review and editing; Model curation. **William T Scott**: Writing—review and editing; Model curation. **Ulf W Liebal**: Writing—review and editing; Model curation. **Lars M Blank**: Writing—review and editing; Model curation. **Hendrik G Mengers**: Writing—review and editing; Model curation. **Mihail Anton**: Writing—review and editing; Model curation. **Albert Tafur Rangel**: Writing—review and editing; Model curation. **Sebastián N Mendoza**: Writing—review and editing; Model curation. **Lixin Zhang**: Writing—review and editing; Model curation. **Jens Nielsen**: Funding acquisition; Writing—review and editing. **Hongzhong Lu**: Conceptualization; Formal analysis; Supervision; Funding acquisition; Writing—original draft. **Eduard J Kerkhoven**: Conceptualization; Formal analysis; Supervision; Funding acquisition; Project administration; Writing—review and editing; Model curation.

Source data underlying figure panels in this paper may have individual authorship assigned. Where available, figure panel/source data authorship is listed in the following database record: biostudies:S-SCDT-10_1038-S44320-024-00060-7.

## Funding

## Disclosure and competing interests statement

The authors declare no competing interests.

# Expanded View Figures

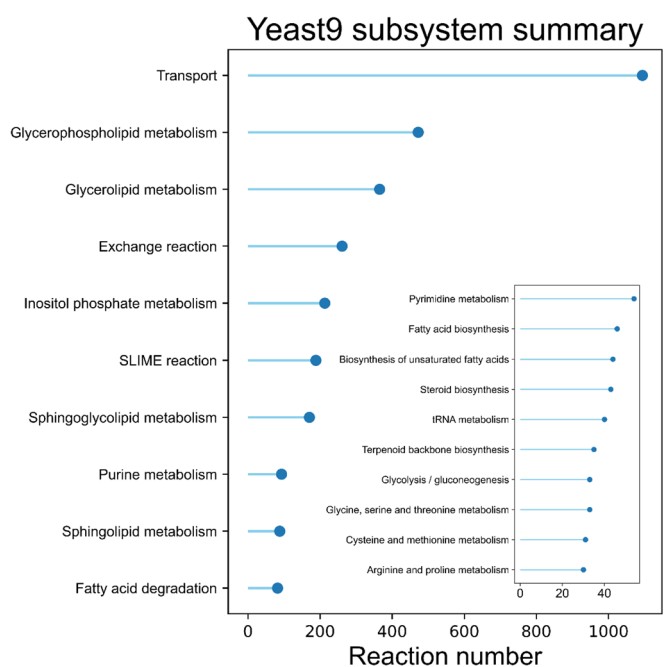

**Figure EV1. Reaction subsystems distribution of yeast-GEM.**

Top 20 reaction subsystems in yeast-GEM.

## Random Forest

| Experiments | | Stress | Unstressed |
|---|---|---|---|
| | Stress | 80 | 0 |
| | Unstressed | 0 | 83 |

Train accuracy: 100%   Test accuracy: 100%

**Figure EV2.  Random Forest classifier prediction performance using transcriptomic data.**

The Random Forest classifier has 100% accuracy in classifying the single cell from high osmotic stress and normal conditions using transcriptomic data.

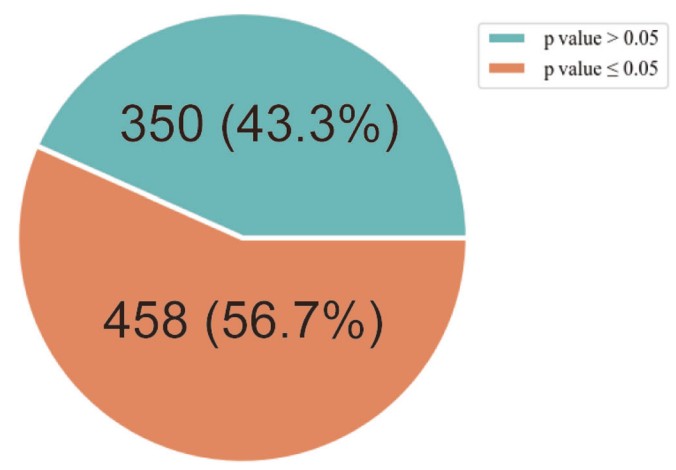

**Figure EV3.  Significance analysis of reactions' flux.**

t test of active flux between salt stress condition and unstress condition.

