## [Peer Review File · Molecular Systems Biology]

Yeast9: A Consensus Genome-scale Metabolic Model for *S. cerevisiae* Curated by the Community

Chengyu Zhang, Benjamín Sánchez, Feiran Li, Cheng Wei Quan Eiden, William Scott, Ulf Liebal, Lars Blank, Hendrik Mengers, Mihail Anton, Albert Rangel, Sebastián Mendoza, Lixin Zhang, Jens Nielsen, Hongzhong Lu, and Eduard J Kerkhoven

Corresponding author(s): Eduard J Kerkhoven (eduardk@chalmers.se)

Review Timeline:

Submission Date:	5th Jan 24
Editorial Decision:	6th Feb 24
Revision Received:	20th May 24
Editorial Decision:	14th Jun 24
Revision Received:	4th Jul 24
Editorial Decision:	8th Jul 24
Revision Received:	17th Jul 24
Accepted:	31st Jul 24

Editor: Maria Polychronidou / Poonam Bheda

Transaction Report:

6th Feb 2024

Manuscript Number: MSB-2024-12208

Title: Yeast9: A Yeast Metabolic Model Enables Quantitative Analysis of Metabolism By Integrating Big Data

Dear Dr. Kerkhoven,

Thank you again for submitting your work to Molecular Systems Biology. We have now heard back from the three reviewers who agreed to evaluate your study. As you will see below, the reviewers raise a series of concerns, which preclude the publication of your study in its current form. Overall, the reviewers think that as it stands the study seems somewhat preliminary and point out that additional analyses and evidence need to be presented to better support the superiority and advantages of Yeast9 over previous models of yeast metabolism.

However, given that the reviewers acknowledge that the presented model can be a relevant contribution for the field of yeast metabolism and metabolic engineering, we have decided to offer you the chance to address the issues raised in a major revision. I think that the reviewers' recommendations are rather clear and straightforward to address. I therefore see no need to repeat any of the comments listed below. All issues raised would need to be satisfactorily addressed. As you may already know, our editorial policy allows in principle a single round of major revision. It is therefore essential to provide responses to the reviewers' comments that are as complete as possible. If you have any questions of if would like to discuss your revision plan with me, please feel free to get in touch.

On a more editorial level, we would ask you to address the following points:

- Please provide a .doc version of the manuscript text (including legends for main figures) and individual production quality figure files for the main Figures (one file per figure).
- We have replaced Supplementary Information by the Expanded View (EV format). In this case (unless the number of EV figures becomes > 6 during revision), all additional figures can be provided as EV Figures. Please provide one file per EV Figure. Their legends should be included in the manuscript text. For detailed instructions regarding expanded view please refer to our Author Guidelines: .
- Supplementary Tables S1-S6 should be provided as EV Tables (for less complex tables, not longer than one page) or EV Datasets (for more complex tables, longer than one page). Please provide one file per EV Table/Dataset. In each file, a description of the table/dataset should be provided in a separate tab.
- Please provide a "standfirst text" summarizing the study in one or two sentences (approximately 250 characters), three to four "bullet points" highlighting the main findings and a "synopsis image" (550px width and max 400px height, jpeg format) to highlight the paper on our homepage.
- All Materials and Methods need to be described in the main text. We would ask you to use 'Structured Methods', our new Materials and Methods format, which is mandatory for Methods and Articles with a strong methodological focus. According to this format, the Material and Methods section should include a Reagents and Tools Table (listing key reagents, experimental models, software and relevant equipment and including their sources and relevant identifiers) followed by a Methods and Protocols section in which we encourage the authors to describe their methods using a step-by-step protocol format with bullet points, to facilitate the adoption of the methodologies across labs. More information on how to adhere to this format as well as downloadable templates (.doc or .xls) for the Reagents and Tools Table can be found in our author guidelines: . An example of a Method paper with Structured Methods can be found here: .
- Please include a Data availability section describing how the data and code have been made available. This section needs to be formatted according to the example below:
The datasets and computer code produced in this study are available in the following databases:
 - Chip-Seq data: Gene Expression Omnibus GSE46748 (<https://www.ncbi.nlm.nih.gov/geo/query/acc.cgi?acc=GSE46748>)
 - Modeling computer scripts: GitHub (<https://github.com/SysBioChalmers/GECKO/releases/tag/v1.0>)
 - [data type]: [full name of the resource] [accession number/identifier] ([doi or URL or identifiers.org/DATABASE:ACCESSION])
- For data quantification: please specify the name of the statistical test used to generate error bars and P values, the number (n) of independent experiments (specify technical or biological replicates) underlying each data point and the test used to calculate p-values in each figure legend. The figure legends should contain a basic description of n, P and the test applied. Graphs must include a description of the bars and the error bars (s.d., s.e.m.).
- Please include a "Disclosure & Competing Interests Statement" in the main text.

- Molecular Systems Biology supports formal data citations in the Reference list, to cite previously published datasets. In addition to citing the original papers that reported the data, we encourage you to also cite the relevant datasets directly in the Reference list. In the text, references to datasets are included as "Data ref: Smith et al, 2001" or "Data ref: NCBI Sequence Read Archive PRJNA342805, 2017". In the Reference list, data citations are very similar to normal literature references but must be labeled with "[DATASET]" at the end of the reference. For detailed instructions please refer to our Author Guidelines .

- When you resubmit your manuscript, please download our CHECKLIST (<https://bit.ly/EMBOPressAuthorChecklist>) and include the completed form in your submission.

Please note that the Author Checklist will be published alongside the paper as part of the transparent process (<https://www.embopress.org/page/journal/17444292/authorguide#transparentprocess>).

If you feel you can satisfactorily deal with these points and those listed by the referees, you may wish to submit a revised version of your manuscript. Please attach a covering letter giving details of the way in which you have handled each of the points raised by the referees. A revised manuscript will be once again subject to review and you probably understand that we can give you no guarantee at this stage that the eventual outcome will be favorable.

Kind regards,

Maria

Maria Polychronidou, PhD
Senior Editor
Molecular Systems Biology

We realize that it is difficult to revise to a specific deadline. In the interest of protecting the conceptual advance provided by the work, we recommend a revision within 3 months (6th May 2024). Please discuss the revision progress ahead of this time with the editor if you require more time to complete the revisions. Use the link below to submit your revision:

IMPORTANT: When you send your revision, we will require the following items:

1. the manuscript text in LaTeX, RTF or MS Word format
2. a letter with a detailed description of the changes made in response to the referees. Please specify clearly the exact places in the text (pages and paragraphs) where each change has been made in response to each specific comment given
3. three to four 'bullet points' highlighting the main findings of your study
4. a short 'blurb' text summarizing in two sentences the study (max. 250 characters)
5. a 'thumbnail image' (550px width and max 400px height, Illustrator, PowerPoint or jpeg format), which can be used as 'visual title' for the synopsis section of your paper.
6. Please include an author contributions statement after the Acknowledgements section (see <https://www.embopress.org/page/journal/17444292/authorguide>)
7. Please complete the CHECKLIST available at (<https://bit.ly/EMBOPressAuthorChecklist>). Please note that the Author Checklist will be published alongside the paper as part of the transparent process (<https://www.embopress.org/page/journal/17444292/authorguide#transparentprocess>).
8. When assembling figures, please refer to our figure preparation guideline in order to ensure proper formatting and readability in print as well as on screen:
<https://bit.ly/EMBOPressFigurePreparationGuideline>
See also figure legend guidelines: <https://www.embopress.org/page/journal/17444292/authorguide#figureformat>
9. Please note that corresponding authors are required to supply an ORCID ID for their name upon submission of a revised manuscript (EMBO Press signed a joint statement to encourage ORCID adoption). (<https://www.embopress.org/page/journal/17444292/authorguide#editorialprocess>)
Currently, our records indicate that the ORCID for your account is 0000-0002-3593-5792.

Link Not Available

*** PLEASE NOTE *** As part of the EMBO Press transparent editorial process initiative (see our Editorial at <https://dx.doi.org/10.1038/msb.2010.72>), Molecular Systems Biology publishes online a Review Process File with each accepted manuscripts. This file will be published in conjunction with your paper and will include the anonymous referee reports, your point-by-point response and all pertinent correspondence relating to the manuscript. If you do NOT want this File to be published, please inform the editorial office at msb@embo.org within 14 days upon receipt of the present letter.

Reviewer #1:

The authors describe the latest updates and improvements for a widely used genome-scale metabolic model (GEM) of *Saccharomyces cerevisiae* that is likely to enable further advances in the field of yeast metabolism research and metabolic engineering. Key updates include the addition of ΔG values for ~98% of metabolites and reaction, addition of reaction subsystems and incremental improvements in reaction coverage and fraction of mass and charge balanced reactions over the previous model (Yeast 8.3.0). Below are suggestions to improve the clarity and accuracy of the manuscript.

1. Line 147: ΔG not defined anywhere before this occurrence. Also, it is important to discuss and clearly state the limitations of ΔG values - dependence on assumptions of concentrations, temperature etc - and that these cannot necessarily be extrapolated to engineered cells or other conditions.
2. In the synthetic lethality predictions (Figure 2c), why are there no true positives ?
3. Transcriptomics integration: What is the accuracy of using just the transcriptomics dataset? The original paper that reported the dataset (Gasch et al. 2017) were able to clearly differentiate the stressed cells from unstressed using transcriptomics. Do the fluxes derived from scGEM improve this separation? Did the authors train a classifier using the transcriptomics dataset alone? Did the authors check what type of distribution of the UMAP transformed data has and whether it still satisfies the assumptions required for training a Naive Bayes classifier?
4. The authors should provide a link to a github repository or more details of the parameters of the naive bayes classifier for reproducibility / model transparency.
5. Flux differences between unstressed and stressed cells based on the scGEM data:
These conclusions seem to be at a population level and presumably, bulk samples of stressed vs unstressed cells would show the same signatures. So it is unclear why single cell data is used. The authors mention the presence of heterogeneity at the flux levels within the single cell populations but do not elaborate on this. The heterogeneity could simply stem from the stochastic nature of the single cell data. A better test dataset would be to explore metabolic flux rewiring in a sample of asynchronous log phase cells that are likely in different phases of the cell cycle.
6. nitrogen limitation simulations: No comparison made with Yeast8. Examples of how the new additions to model enable or improve this analysis are missing.
7. Conclusion on weak correlation between proteomes and fluxes: The authors should provide a summary of how many fluxes correlated with protein abundance levels for both the nitrogen source utilisation simulations as well as the phenotype constrained simulations. In lines 258-265, they exemplify reactions for which flux and protein concentration values agree while in the analysis described in Lines 266-278, they conclude that protein and flux level changes are weakly correlated.

If, in general, protein and flux levels do not correlate, are the consistent fluctuations observed for the first set of simulations (nitrogen source utilisation) an exception?

If so, perhaps the authors can elaborate on the metabolic subsystems for which protein and flux bear high correlation vs those for which they do not, and speculate on why this occurs.

The authors use the phrase "transcriptionally regulated" for referring to protein level changes measured by proteomics. Observations from proteomics data should be treated as "protein abundance level" and those from RNAsequencing data as "transcript level" or "regulated at the RNA level" as protein abundances as controlled both by transcriptional and post-translational regulatory mechanisms.

8. Growth profiles and gene function prediction using T'omics:

To demonstrate the usability of Yeast9 for functional annotation of genes, the authors created single strain GEMs for single or double knockout strains and use transcriptomics measurements from these strains to constrain each model (by altering reaction bounds). They simulated fluxes using FBA for each knockout and obtained a high correlation between the measured and simulated growth rates of the strains. They use 16 functional bioprocess categories also used by others ~10 years ago to label the functions of all the deleted genes. The simulated fluxes were then processed using kernel PCA (kPCA) to reduce the dimensionality of the data before being used as input for training multiple machine learning models for classifying each knockout into one of the 16 functional categories.

1) GEMs are built using gene annotations, and simulations are based on T'omics data, thus there is a degree of circularity in predicting functions and comparing these with T'omics only predictions.

2) the functional categories they use as labels seem to be outdated (all but one of the 5 studies they cite in Lines 307-308 are ~10 years old and the one study that was published in 2020 was focused on predicting growth of KOs rather than gene function).

- Supplementary File 5 has categories that are not evenly distributed across cell functions, two categories of the 16 are for unclear / unknown functions (corresponding to 225/1485 total genes or 15% of the dataset). Many genes in the "other" category do have known functional annotations eg CCE1, CEX1, PHO12, PHO81, PHO84, GRX3, GRX4
Much better annotations available - even in just GOSlim or GO. We recommend evaluating the protein function prediction using community standard used by others working on ML based protein function prediction such as those summarised in (Radivojac et al. 2013) (includes *S. cerevisiae* data)
- Genes can have multiple functions and multiple types of functional annotations. Most other tools that predict gene function output multiple class labels for each gene. Why did the authors choose to predict only one label per gene?

Example of a gene that is classified as "hypothetical/unknown" by the authors labels but has something known about its function on UniProt: Vps13

- classified as hypothetical/unknown function by the GEM based prediction
- described on UniProt as mediating transfer of lipids between membranes at organelle contact sites
<https://www.uniprot.org/uniprotkb/Q07878/entry#sequences>
- DeepFRI based prediction : GO CC - membrane bound organelle, ATP binding and many other terms.

Other minor concerns:

- No comparison of growth rate predictions to other features that can be used has been made eg. transcriptome alone (Culley et al. 2020) , proteome (eg (Messner et al. 2023) report an $R^2 = 0.68$ based on a random forest model).
- How many of the KOs correspond to genes that are part of the GEM ?
- Does the prediction accuracy of knockout strains differ based on whether the KOs are part of the GEM?
- Are there particular categories of functional annotations that are predicted better than others?
- Can the authors classify any previously uncharacterised gene into a meaningful functional category using this methodology?

Questions about technical details:

- Were 500 features used in each case for flux only, transcriptomics only, flux and transcriptomics?
- Which kernel was used in the kPCA?
- What criteria were used to optimise the parameters/kernel used?
- Is the same type of kernel ideal for transforming both the transcriptomics and flux data?
- Why is the data processing pipeline for this ML prediction different from the one used for the single cell GEM fluxes?
Perhaps the authors could comment on considerations or any exploratory data analysis that led them to use UMAP for the single cell data and kPCA for the knockout dataset and why one approach is better than the other for future applications for GEM fluxes as input for ML models.
- Limited description of the ML models, their strengths and limitations, the parameters used for each model and any hyperparameter tuning or optimisation that was conducted. Perhaps the authors could adopt the model cards approach (<https://modelcards.withgoogle.com/about>) or a similar documentation strategy being used within the ML community to provide more transparency (at least for their best performing GEM based model).
- How generalisable is the model? Is there any other published transcriptomics dataset from an independent experiment that could be used to conduct cross-validation?

Culley, Christopher, Supreeta Vijayakumar, Guido Zampieri, and Claudio Angione. 2020. "A Mechanism-Aware and Multiomic Machine-Learning Pipeline Characterizes Yeast Cell Growth." *Proceedings of the National Academy of Sciences of the United States of America* 117 (31): 18869-79.

Gasch, Audrey P., Feiqiao Brian Yu, James Hose, Leah E. Escalante, Mike Place, Rhonda Bacher, Jad Kanbar, et al. 2017. "Single-Cell RNA Sequencing Reveals Intrinsic and Extrinsic Regulatory Heterogeneity in Yeast Responding to Stress." *PLoS Biology* 15 (12): e2004050.

Gligorijević, Vladimir, P. Douglas Renfrew, Tomasz Kosciolk, Julia Koehler Leman, Daniel Berenberg, Tommi Vatanen, Chris Chandler, et al. 2021. "Structure-Based Protein Function Prediction Using Graph Convolutional Networks." *Nature Communications* 12 (1): 3168.

Messner, Christoph B., Vadim Demichev, Julia Muenzner, Simran K. Aulakh, Natalie Barthel, Annika Röhl, Lucía Herrera-Domínguez, et al. 2023. "The Proteomic Landscape of Genome-Wide Genetic Perturbations." *Cell* 186 (9): 2018-34.e21.

Radivojac, Predrag, Wyatt T. Clark, Tal Ronnen Oron, Alexandra M. Schnoes, Tobias Wittkop, Artem Sokolov, Kiley Graim, et al. 2013. "A Large-Scale Evaluation of Computational Protein Function Prediction." *Nature Methods* 10 (3): 221-27.

Reviewer #2:

This manuscript describes the release and test of the newest version of the Yeast GEM, version 9. The subject is timely. The manuscript is very well-written, the study is well designed and conducted, and the obtained results are very interesting.

The following issues should be addressed to improve the quality of the manuscript:

- Line 104 and similar quotes - the authors talk frequently about "condition-specific GEMs (csGEMs)" or "strain-specific GEMs (ssGEMs)"; although I get the idea, that means that we need a different model for every condition, so we need omics data every time we want to change the condition in which we operate. Does that not mean that model alone does not work? At least please discuss the issue.
- Line 173 - should the case described here be solved by the integration of transcriptomics data, given that the impact of redundant gene expression would be accounted for?
- Line 199 - describe how relative transcript levels were converted into constraints of the model.
- Line 353 - discuss whether the massive number of simulations helped to improve the quality of the underlying model, irrespective of the specific conditions tested.

Minor issues:

Line 59 - Goffeau's first paper describing the yeast genome seq is dated 1996: Goffeau A, et al. (1996) Life with 6000 genes. Science 274(5287):546, 563-7.

Line 83 - Delete "is"

Figure 2e - clarify what was the criteria used to select the tested C and N sources?

Reviewer #3:

General comments

The authors present the latest iteration of the consensus yeast genome-scale model (Yeast-GEM 9) and illustrate its utility through various case studies. There have been significant updates since the previous version (Yeast GEM 8). Metabolic modeling of yeast is useful for a broad spectrum of research areas ranging from industrial applications to fundamental studies of metabolic processes and their regulatory mechanisms. The importance of this work is unequivocally evident. However, despite our inclination to recommend this study for publication, there is a major issue concerning its presentation that prevents us from doing so. Starting with the title, the manuscript implies that Yeast GEM 9 has unique capabilities, but fails to show this with a comparison to other yeast models, except for a couple panels in Figure 2. In fact, we believe most demonstrated capabilities are inherent in Yeast GEM 8 (as well as in any good quality model, I should add). Addressing this issue is crucial, either through direct comparisons with Yeast 8 to highlight the advancements or by revising the narrative to reflect the true scope of the model's improvements.

Regardless, there are also several areas of concern that need to be addressed to ensure the study's overall quality (please see major comments below).

Major Comments (in the order of appearance in the paper):

1) We like the idea of nearly comprehensive deltaGo annotations, which is one example of novel capabilities in Yeast GEM 9. However, I have multiple concerns with the way this addition was handled. Firstly, deltaGo of metabolites are collected from multiple sources. Are these consistent with each other (for the shared metabolites)? Why not use dGpredictor, for example, throughout all metabolites, so that all values are obtained with the same rules? Secondly, the reason some metabolites are missing in the yETFL model seems to be because this resource refused to provide deltaGo values for membrane-associated metabolites, as corrections are needed but unknown for non-aqueous microenvironments. How is this concern addressed here? Thirdly, it is hard to access the deltaGo data (I see it only in .mat model, not in other models like the sbml version, and I also see it in csv files but these are to be searched in github folders), and this data that I found does not indicate the source of the values. And finally, the analysis presented with deltaGos (mapping the values to classical central carbon pathways) is superficial. I thought we could already do this analysis, for example, using MetaCyc pathways and deltaGo values therein. The utility of the deltaGo addition to Yeast GEM is therefore not clear.

2) The result of the synthetic lethality analysis is presented as "Synthetic lethality can be predicted with near-perfect accuracy (Fig. 2c), whereas a few cases of false negative predictions still exist (Supplementary File 2)." This appears to be an overstatement. In fact, since the model predicted no positives, we cannot even talk about recall or precision, because of which I could argue that the model predictions do not have any value. Unless we are missing something, Figure 2c implies that predictions of this model are equivalent to saying "there are no synthetic lethal gene pairs in yeast", which is not very meaningful but is as accurate as the presented predictive algorithm. The authors suggest that the inability to predict positives is partially due

to apparent redundancies, which we agree, but then we suggest a different gene deletion algorithm which removes all reactions of a deleted gene no matter if other genes are associated with those reactions and how. How would predictive ability change then? How many of the experimental positives would be predicted?

3) Flux predictions based on the integration of single-cell data with GIMME algorithm are used to classify cells in two groups. How can we evaluate the predictive power of this classification without a reference point? One possibility is establishing a baseline by doing the same classification with gene expression alone (another possibility is comparing the results with those from Yeast GEM 8, but this general criticism is applicable to pretty much any case study [see above in general comments]). In addition, we do not believe we can yet reach conclusions from this modeling analysis as in "In terms of evolution, heterogeneity at the single-cell level may assist *S. cerevisiae* populations to efficiently adapt to new environments", because the heterogeneity in predicted flux distributions may be an artifact of sparse single-cell data, or any other factor, which may cause inaccurate predictions by the GIMME algorithm. However, some additional steps, such as randomization of cell labels and rerunning the classification n times, to establish the statistical significance of predictions may be taken to reach such conclusions.

4) The calculation of the preference score used in the evaluation of nitrogen sources require some clarification and more detailed explanations than what is provided. This method and data obtained from it are counter intuitive. In particular, how can glucose, which has no nitrogen atoms, compensate for the limitation of a nitrogen source?

5) The authors reach a strong conclusion that the correlation between protein abundance and flux is low, based on fluxes predicted from chemostat data (Figure 4d). Without understanding the uncertainty in the predicted flux values, this conclusion may be premature. Flux variability analysis, or flux sampling as the authors seem to have done, can be used to assess how tight the flux values are, given the experimental constraints. If the variability is generally high, the lack of correlation may also be explained by our low confidence in the predicted flux values, meaning the conclusion is not valid. The related section in Methods "Simplified regulation analysis based on sampled fluxes and absolute proteomics" is unclear and poorly written.

6) We have several concerns also with the GEM reconstructions for knock-out strains: From the related methods section, where the explanations are not clear, the integration of gene expression data seems to be done by setting flux boundaries as gene expression levels (although how upper and lower fluxes are set differently is also not clear). What is the justification for this? Since the method is also not readily comprehensible in the referred studies, some explanation (or correction in the methods) seems to be necessary. In addition, the referred study from Culley et al. (2020) reports the same level of correlation (0.66) between predicted and measured growth rates as this study found (Fig 5b). Please comment on this. Finally, for a full comparison between transcriptomics-based and fluxomics-based predictions from machine learning, we would need to see the train set/test set accuracy table, as in Table 1, for the transcriptomics-based predictions (Table 1 is for fluxomics-based predictions only). For example, random forests classifier seems like the best predictor with fluxomics in the figure (5c), but is rejected because of poor test set accuracy, which is not shown in the figure but available in the table for fluxomics-based models. There is no such table for transcriptomics-based models.

Minor comments and suggestions:

1) It would be nice to provide a list of modifications in Yeast GEM 9 compared to version 8 (additions and subtractions as reactions, genes, and metabolites) with notes about the reason for each modification.

2) As the authors also indicate, ΔG (standard Gibbs free energy change) values are not sufficient to determine reaction reversibility. Actual ΔG values are also a function of metabolite activity coefficients. Is there a database from which probable ranges of activity for most yeast metabolites can be obtained, so that we can see a range of ΔG (min and max) in addition to ΔG annotations? A range not crossing 0 would then indicate likely irreversibility in one direction. This would be very useful and unique.

3) The title of the last section of Results "Growth profiles and gene function can be predicted by Yeast9 constrained with large scale transcriptomic" seems truncated.

4) The data in Table 1 indicates that random forest (RF) classifier was overfitted (test score is much lower than training score which is itself nearly perfect). However, RF models are especially resilient against overfitting. This raises the question if the modeling parameters are optimized properly for all models. In the RF example, did the authors use out-of-bag scores during model training to adjust the parameters (number of trees, tree depth etc.) appropriately? More broadly, the methodology for ensuring a fair and balanced comparison across the different models used in this analysis needs clarification.

5) Line 391: How is protein localization predicted? I am sure this can be found in previous publications, but it is worth mentioning briefly.

6) Methods section titled "Condition-specific GEMs construction and simulation under nitrogen limitation" is confusing. The first paragraph says the model is constrained with experimental growth rate, but the first step indicates a specific (constant) constraint at 0.1 1/h.

7) Figure 4c: flux values are hard to visualize. Please sort from large to small (or the other way around) or use small bar graphs instead of small tables.

Reviewer #1:

The authors describe the latest updates and improvements for a widely used genome-scale metabolic model (GEM) of *Saccharomyces cerevisiae* that is likely to enable further advances in the field of yeast metabolism research and metabolic engineering. Key updates include the addition of ΔG values for ~98% of metabolites and reaction, addition of reaction subsystems and incremental improvements in reaction coverage and fraction of mass and charge balanced reactions over the previous model (Yeast 8.3.0). Below are suggestions to improve the clarity and accuracy of the manuscript.

Response: We thank the reviewer for the kind and constructive comments. In the revised manuscript, as suggested, we systematically optimized the machine learning parameters, evaluated various data preprocessing methods, provided detailed descriptions of the machine learning approaches, provided updated synthetic lethality predictions and discussed ΔG . We further rephrased text and improve the storyline.

1. Line 147: ΔG ' not defined anywhere before this occurrence. Also, it is important to discuss and clearly state the limitations of ΔG values - dependence on assumptions of concentrations, temperature etc - and that these cannot necessarily be extrapolated to engineered cells or other conditions.

Response: We agree with the reviewer that the meaning of ΔG° warrants more detailed explanation, including a discussion of its limitations. We have now modified this in the text (lines 161-167 and 173-179) as:

The ΔG° value is indicative of the thermodynamic driving force of a reaction under biochemical standard conditions, i.e. pH 7, 298 K, 1 atm and unit concentrations for chemicals other than water and protons. Nonetheless, as reaction directionality can be determined by calculating condition-specific in vivo ΔG values with $\Delta G = \Delta G^\circ + RT \ln Q$ (where Q is the ratio of product of reactant concentrations), the ΔG° is still somewhat indicative of the likelihood of reaction reversibility.

[...]

In central carbon metabolism (Figure 2F), pathways such as glycolysis, tricarboxylic acid cycle and pentose phosphate pathway have negative the ΔG° with -22.8 , -9.6 and -13.8 kJ/mol, respectively. The ΔG° of individual reactions, however, can range drastically, implying that metabolites reach concentrations (i.e. Q) that are compatible with a forward flux through these reactions. This exemplifies the importance of considering metabolite concentrations before drawing conclusions on the effect of ΔG° on the functioning of the metabolic network.

2. In the synthetic lethality predictions (Figure 2c), why are there no true positives ?

Response: We are very grateful that multiple reviewers commented on this, as the previously reported result suffered from a bug in our analysis. We have corrected this, and in our updated analysis there are 6506 true positives, giving an overall accuracy of almost 80% (Fig R1). This has resulted in an updated Figure 2 and discussion of these results in lines 151-159).

		Model simulations	
		Synthetic lethal	Not synthetic lethal
Experiments	Synthetic lethal	True positive 6506	False negative 43293
	Not synthetic lethal	False positive 58860	True negative 419158

Figure R1 The Yeast9 could predict the consequences of synthetic lethal of two gene combinations, with the accuracy at 80%.

3. Transcriptomics integration: What is the accuracy of using just the transcriptomics dataset? The original paper that reported the dataset (Gasch et al. 2017) were able to clearly differentiate the stressed cells from unstressed using transcriptomics. Do the fluxes derived from scGEM improve this separation? Did the authors train a classifier using the transcriptomics dataset alone?

Did the authors check what type of distribution of the UMAP transformed data has and whether it still satisfies the assumptions required for training a Naive Bayes classifier?

Response: We are grateful for the constructive feedback. As mentioned, the original paper (Gasch et al. 2017) was able to classify cells from whole transcriptomics datasets, while in our approach we analyze whether the metabolic flux perspective would be sufficient to classify stressed from unstressed cells. A fundamental assumption of the Naive Bayes classifier that we previously used is the independence among features. Upon closer examining of the data processed by UMAP, we observed that the p-value was around 0.2, which does not meet this assumption. Consequently, we selected the Random Forest model due to its less stringent requirements for data feature independence.

We conducted a comparative analysis using transcriptome and fluxome datasets as inputs for the model training. The data preprocessing was categorized into three approaches: no preprocessing, dimensionality reduction using UMAP, and kernel PCA (kPCA). During the model training phase, we employed scikit-learn's GridSearchCV to fine-tune the parameters, which included the number of estimators (n_estimators), maximum depth (max_depth), minimum samples for a split (min_samples_split), and minimum samples for a leaf (min_samples_leaf). If kPCA was used for data preprocessing, the number of components (n_components) and the kernel in kPCA were also considered in the parameter optimization.

The outcomes are presented in Table R1, and the best parameters are listed in Table R2. While the UMAP derived results can be discredited for not satisfying the feature independence requirement, we observed that kPCA transformed data performed better than having no pre-

processing (and better than our earlier presented UMAP-based results). Comparing fluxomics with transcriptome shows consistent higher accuracy for the latter. This is likely due to the transcriptomics provides more data, and while non-metabolic genes (that are therefore not considered in the fluxomics data) might contribute to the higher accuracy of transcriptomics data, it is apparent that the fluxomics also captures a large part of the differential phenotypes. We have updated Fig 3b and now report on these results in lines 196-206 as:

Kernel principal component analysis (kPCA) was executed to extract features before machine learning. The **random forest** classification was trained using the **kPCA**-processed data. **After parameter optimization, the random forest** classifier demonstrated **78%** accuracy in differentiating single cell sampled from the osmotic stress and unstressed conditions (Fig. 3). **The optimized parameters are presented in Table EV2 .** **When the same analysis on the whole transcriptomics dataset yielded 100% accuracy (Figure EV2), corroborating the observations by Gasch *et al.* (2017), this implied that the metabolic networks of individual cells were sufficiently different to enable categorizing them by which environment they resided **without considering differential expression of non-metabolic gene.** However, differential expression of non-metabolic genes are important determinants of the osmotic stress state that likely have strong biological relevance.**

As transcriptomics already reaches 100% accuracy, it is meaningless to evaluate the accuracy when combining transcriptomics and fluxomics, as was done later in the manuscript.

Table R1 The Random Forest simulation.

Data preprocess	Data type	Test data accuracy	Train data accuracy	True positive	False positive	True negative	False negative
kPCA	flux	0.78	1	79	1	73	10
	trans	1	1	80	0	83	0
umap	flux	0.71	0.88	66	14	69	14
	trans	1	1	80	0	83	0
No	flux	0.76	1	74	6	77	6
	trans	1	1	80	0	83	0

Table R2 Best parameters of the Random Forest simulation.

	rf_flux_No	rf_flux_umap	rf_flux_kPCA	rf_trans_No	rf_trans_umap	rf_trans_kPCA
rf_max_depth	None	None	None	None	None	None
rf_min_samples_leaf	1	4	1	1	1	1
rf_min_samples_split	2	2	2	2	2	2
rf_n_estimators	100	10	100	10	100	10
kPCA_kernel			linear			linear

4. The authors should provide a link to a github repository or more details of the parameters of the naive bayes classifier for reproducibility / model transparency.

Response: All codes used in this paper are stored in

https://github.com/hongzhonglu/yeast_GEM_multi_omics_analysis, to which we refer in the Data availability section.

5. Flux differences between unstressed and stressed cells based on the scGEM data:

These conclusions seem to be at a population level and presumably, bulk samples of stressed vs unstressed cells would show the same signatures. So it is unclear why single cell data is used. The authors mention the presence of heterogeneity at the flux levels within the single cell populations but do not elaborate on this. The heterogeneity could simply stem from the stochastic nature of the single cell data. A better test dataset would be to explore metabolic flux rewiring in a sample of asynchronous log phase cells that are likely in different phases of the cell cycle.

Response: The aim of our manuscript is the introduction of the latest release of yeast-GEM and the demonstration of its capability and significance in integrating big data. The reviewer is correct that single cell data might not be essential to study osmostress, and we agree that the proposed dataset of asynchronous log phase cells might be more suitable in single cell analysis. However, our rationale has not been to investigate metabolism and osmostress “with the best dataset and approach”, but rather “can single-cell data and GEMs be combined to represent single-cell networks”. We are confident that we demonstrate just that. In lines 183-185, we have clarified the rationale for this analysis as following:

While yeast-GEM represents the theoretical metabolic network of *S. cerevisiae* based on its genome annotation, not all enzymes might be constitutively expressed. **Gene expression may differ between cells, and consequentially the metabolic network of individual cells may not be the same.** To examine this, we collected 163 single-cell transcriptomes of *S. cerevisiae* (Gasch *et al*, 2017) including 80 transcriptomes measured under high osmotic stress and 83 transcriptomes measured under reference conditions.

And we mention in lines 187-192 that the difference between scGEMs might indeed come from stochasticity of the single-cell approach, as well as true differences in gene expression:

We used GIMME (Becker & Palsson, 2008) to construct 163 single-cell omics-constrained GEMs (scGEMs) **by modify the presence of reactions and metabolites in the model based on transcriptomic data (Fig. 3A).** Strictly, due to the nature of single-cell data, it is not known whether the scGEMs reflect true inter-cell variability, or rather only reflect stochasticity in the single-cell data acquisition, as is a challenge in any single-cell approach.

6. nitrogen limitation simulations: No comparison made with Yeast8. Examples of how the new additions to model enable or improve this analysis are missing.

Response: The comparison of Yeast8 with Yeast9 is described in the beginning of the Results section, visualized in Figure 1-2 and all commits in between:

<https://github.com/SysBioChalmers/yeast-GEM/compare/v8.3.0...v9.0.0>, where numerous metrics exemplify the progress made between model versions, covering both improved performance (e.g. gene essentiality prediction) and increased functionality (e.g. providing ΔG° values). After systematically evaluating the improvements of Yeast9, we proceed with further analyses where our intention has not been to remain focused on evaluating Yeast8 against Yeast9. Rather, we aimed to provide novelty to various recent published studies, through integrative analysis of the respective data with Yeast9, thereby demonstrating the value using Yeast9. Similar storylines have repeatedly been followed when announcing updates of other widely-used genome-scale models (Monk *et al*, 2017; Lu *et al*, 2019; Robinson *et al*, 2020). Throughout the manuscript we have made adjustments to reflect this, such as referring to yeast-GEM instead of Yeast9 as being suitable for a particular type of analysis (e.g. lines 113, 118, 276, 305).

7. Conclusion on weak correlation between proteomes and fluxes: The authors should provide a summary of how many fluxes correlated with protein abundance levels for both the nitrogen source utilisation simulations as well as the phenotype constrained simulations. In lines 258-265, they exemplify reactions for which flux and protein concentration values agree while in the analysis described in Lines 266-278, they conclude that protein and flux level changes are weakly correlated.

If, in general, protein and flux levels do not correlate, are the consistent fluctuations observed for the first set of simulations (nitrogen source utilisation) an exception?

If so, perhaps the authors can elaborate on the metabolic subsystems for which protein and flux bear high correlation vs those for which they do not, and speculate on why this occurs.

The authors use the phrase "transcriptionally regulated" for referring to protein level changes measured by proteomics. Observations from proteomics data should be treated as "protein abundance level" and those from RNA sequencing data as "transcript level" or "regulated at the RNA level" as protein abundances as controlled both by transcriptional and post-translational regulatory mechanisms.

Response: From the reviewer's comment it appears as if the manuscript was ambiguous on the observed correlation between protein and flux levels. It is critical to note that in the manuscript, we first compared non-preferred nitrogen sources (isoleucine and phenylalanine) with ammonia, in otherwise identical conditions: this involved two cross-comparisons. There, around 160 genes showed a consistent directionality in their response to the change in nitrogen source: e.g. they were significantly up on both the proteome and flux level. Of those, 149 genes were shared for both non-preferred nitrogen sources, enriched for amino acid biosynthesis. We have clarified this in lines 254-259 as follows:

Next, we analyzed consistent tendencies (i.e. not correlations, but whether directions of change agreed) between reaction fluxes and the related protein abundances, comparing the unpreferred nitrogen sources (isoleucine or phenylalanine) with the favored one (i.e., ammonia). In these two cross-comparisons, 164 (isoleucine) and 166 (phenylalanine) of the Yeast9 genes show consistent directional change in both flux and protein level (Table EV3).

Then, we determined the covariance more precisely between fluxes and proteomics across diverse environments. For this, we expanded our analysis by using 12 phenotype-constrained models (not only changing nitrogen source, but also dilution rate and C:N ratio). In this second analysis, we evaluated flux-protein pairs that showed a correlated and proportional response across all of these conditions by calculating ρ values. We aimed to visualize this in Figure 4D by a schematic graph where a correlation is indicated for multiple flux and protein datapoints. Here, only 11 reactions showed comparatively high correlation with $\rho > 0.5$ (albeit still far off from the optimal value $\rho = 1$). As such, the earlier identified 160 genes that responded to nitrogen utilization were not an exception, they were determined in a different type of analysis. Consistent directionality of change when comparing 2 conditions is much simpler to accomplish in comparison with linear correlation across 12 conditions where 3 different parameters are varied.

The combination of these two analyses showed that (a) the direction of differential protein expression as a response to environmental changes can show correlation with changes in metabolic flux for some reactions (here, changing nitrogen source affected amino acid biosynthesis); but (b) the protein expression values across multiple conditions correlate poorly with metabolic fluxes values. While up- or downregulation of protein levels can direct increased or decreased fluxes, the size of the differential expression is not directly proportional to the flux change. It is likely that this is due to additional post-translational regulatory mechanisms. We have clarified this in lines 263-268 and 275-277 as follows:

To more precisely determine the covariance between fluxomics and proteomics across diverse environments, we expanded the analysis to not only consider the direction of change in both flux and protein level, but to evaluate if protein and flux levels quantitatively correlated across multiple conditions. We thereby utilized 12 phenotype-constrained models (Fig. 4D) and their associated proteomic datasets, covering not only alternative nitrogen sources, but also six dilution rates and three carbon-nitrogen ratios, in a regulatory analysis that has previously been described for *E. coli* (Kochanowski et al, 2021).

[...]

This implies a low correlation between protein abundance and flux, albeit the direction of change might be better conserved. Additional post-translational regulatory mechanisms are likely implicated in this discrepancy.

In addition, we have corrected our use of the phrase “transcriptional regulation”, per the valuable comments of the reviewer.

8. Growth profiles and gene function prediction using T'omics:

To demonstrate the usability of Yeast9 for functional annotation of genes, the authors created single strain GEMs for single or double knockout strains and use transcriptomics measurements from these strains to constrain each model (by altering reaction bounds). They simulated fluxes using FBA for each knockout and obtained a high correlation between the measured and simulated growth rates of the strains. They use 16 functional bioprocess categories also used by others ~10 years ago to label the functions of all the deleted genes. The simulated fluxes were then processed using kernel PCA (kPCA) to reduce the dimensionality of the data before being used as input for training multiple machine learning models for classifying each knockout into one of the 16 functional categories.

1) GEMs are built using gene annotations, and simulations are based on T'omics data, thus there is a degree of circularity in predicting functions and comparing these with T'omis only predictions.

2) the functional categories they use as labels seem to be outdated (all but one of the 5 studies they cite in Lines 307-308 are ~10 years old and the one study that was published in 2020 was focused on predicting growth of KOs rather than gene function).

- Supplementary File 5 has categories that are not evenly distributed across cell functions, two categories of the 16 are for unclear / unknown functions (corresponding to 225/1485 total genes or 15% of the dataset). Many genes in the "other" category do have known functional annotations eg CCE1, CEX1, PHO12, PHO81, PHO84, GRX3, GRX4

Much better annotations available - even in just GOslim or GO. We recommend evaluating the protein function prediction using community standard used by others working on ML based protein function prediction such as those summarised in (Radivojac et al. 2013) (includes *S. cerevisiae* data)

- Genes can have multiple functions and multiple types of functional annotations. Most other tools that predict gene function output multiple class labels for each gene. Why did the authors choose to predict only one label per gene?

Example of a gene that is classified as "hypothetical/unknown" by the authors labels but has something known about its function on UniProt: Vps13

- classified as hypothetical/unknown function by the GEM based prediction

- described on UniProt as mediating transfer of lipids between membranes at organelle contact sites <https://www.uniprot.org/uniprotkb/Q07878/entry#sequences>

- DeepFRI based prediction : GO CC - membrane bound organelle, ATP binding and many other terms.

Response: Regarding the first comment, the potential circularity between function prediction and simulation with transcriptomics data is not a major concern (although it cannot be completely excluded) when we consider the analysis pipeline. It is true that the starting GEM (i.e. Yeast9) is based on functional annotation: the assignment of reactions to enzyme-coding genes. However, the transcriptomics data from which ssGEMs are subsequently derived is

obtained from knockouts of genes with a large variety of functions. Of those, only 75 genes are in the model, and 11 of these genes are assigned to the five classes that we aim to identify.

Moreover, simulations with the ssGEMs aim to represent the metabolic capabilities after each knockout, and our analysis shows that knockout of genes that have functions in the same bioprocess category give similar changes in predicted fluxes. We do not aim to predict precise enzyme functionality, in which case there would have been a direct circularity problem. Moreover, we do not use the simulated fluxes directly, but we rather use them in machine learning for classifying genes into bioprocess categories. The results indicate that the predictive capabilities of transcriptomics and flux vary across different machine learning models, with their combined use yielding optimal predictive performance (Table R3, in the manuscript as Table 1). This suggests that transcriptomics and flux data may be applicable in different contexts and can serve as complementary to one another.

Out of curiosity, we have also run the ML analysis while leaving out knockouts of the 11 genes that are assigned in yeast-GEM. This did not drastically change the performance (cf. Table R3 and R4), and we therefore decided not to include this in the manuscript.

Regarding the second comment, single-class assignments are imperative for our envisioned analysis pipeline. From a machine learning standpoint, the assignment of more than one function per gene (as is e.g. observed with GO terms) would turn it into a significantly more complex problem, which we purposely decided to avoid. The reviewer raises a few valuable suggestions for functional annotations, but these all assign multiple functions per gene. We therefore settled for the five functional categories from Culley *et al.* (2020) that had the highest number of genes: cell cycle regulation, chromatin factors, gene-specific transcription factors, protein kinases, and ubiquitin(-like) modifications. The “other” category was not utilized in the machine learning approach. We explored four other sources of functional categories with single-class assignments, but decided not to use those as the group sizes were too small to be conducive for machine learning. As the reviewer commented, these four sources were all over 10 years old, and the reviewer raises additional arguments why these would not be ideal assignments to consider. We already did not consider them in our analysis, and to avoid confusion we removed any mention of them from the manuscript.

We have now significantly rewritten the text in the revised manuscript, see lines 305-327 and updated Table 1. All the genes used for machine learning classification (totaling 559, spread across five categories) are provided in the Table EV7.

Table R3 Profile of machine learning simulation.

Dataset	Algorithm	Train data accuracy	Test data accuracy	Test data recall rate	Test data F score
Transcriptomic	SVM	0.99	0.68	0.66	0.67
	MLP	1.00	0.67	0.66	0.66
	Naïve Bayes	0.58	0.32	0.35	0.30
	Random forest	1.00	0.57	0.57	0.56
	k-NN	1.00	0.60	0.58	0.60

	Logistic	0.99	0.72	0.70	0.70
	SVM	0.52	0.39	0.35	0.33
	MLP	0.34	0.31	0.27	0.17
Flux	Naïve Bayes	0.92	0.88	0.89	0.88
	Random forest	0.99	0.70	0.69	0.70
	k-NN	1.00	0.57	0.54	0.55
	Logistic	0.47	0.32	0.28	0.24
	SVM	0.84	0.53	0.51	0.51
Transcriptomic and Flux	MLP	0.28	0.30	0.30	0.28
	Naïve Bayes	0.94	0.90	0.90	0.90
	Random forest	1.00	0.83	0.82	0.83
	k-NN	1.00	0.43	0.41	0.41
	Logistic	0.55	0.37	0.36	0.36

Table R4 Profile of machine learning simulation, excluding 11 yeast-GEM associated genes from the dataset.

Dataset	Algorithm	Train data accuracy	Test data accuracy	Test data recall rate	Test data F score
Transcriptomic	SVM	0.99	0.65	0.63	0.63
	MLP	1.00	0.70	0.67	0.68
	Naïve Bayes	0.60	0.38	0.40	0.37
	Random forest	1.00	0.61	0.58	0.58
	k-NN	1.00	0.61	0.59	0.59
	Logistic	1.00	0.67	0.66	0.66
Flux	SVM	0.81	0.36	0.32	0.31
	MLP	0.34	0.35	0.29	0.19
	Naïve Bayes	0.93	0.87	0.87	0.87
	Random forest	1.00	0.87	0.85	0.86
	k-NN	1.00	0.45	0.44	0.45
	Logistic	0.63	0.33	0.34	0.32
Transcriptomic and Flux	SVM	0.63	0.33	0.34	0.32
	MLP	1.00	0.43	0.44	0.43
	Naïve Bayes	0.92	0.82	0.82	0.83
	Random forest	1.00	0.74	0.73	0.73
	k-NN	0.37	0.35	0.31	0.24
	Logistic	1.00	0.35	0.32	0.29

Other minor concerns:

- No comparison of growth rate predictions to other features that can be used has been made eg. transcriptome alone (Culley et al. 2020) , proteome (eg (Messner et al. 2023) report an R2 = 0.68 based on a random forest model).

Response: Based on the Reviewer's comments, we have now included this importance comparison of our growth rate predictions with those obtained by Culley et al. (2020) on transcriptomics only, which is a fair comparison as they used the same dataset. See lines 295-

304.

Flux balance analysis (FBA) with growth maximization as the objective function showed a good correlation between the predicted and measured growth rate, with Pearson correlation coefficient (PCC) = 0.66 for single knockout strains and 0.78 for double knockout strains (Fig. 5B). [...] Meanwhile, it is not only the expression of metabolic genes (and their corresponding constraints on the metabolic model) that dictate strain specific growth rates, as growth rate predictions based solely on the whole transcriptomics dataset reached $PCC > 0.9$ (Culley *et al*, 2020).

- How many of the KOs correspond to genes that are part of the GEM ?

Response: There are 75 knock-out genes that are part of the GEM, we have now explicitly mentioned this in lines 291-293:

Of these 1,143 knocked out genes, 75 were assigned to reactions in Yeast9, implying that the ssGEMs are mostly reflecting the metabolic networks after knockout of non-enzyme-coding genes.

- Does the prediction accuracy of knockout strains differ based on whether the KOs are part of the GEM?

Response: When separately evaluating the growth rate prediction accuracy with the 75 knockout genes, a PCC of 0.67 was found, which is comparable with the PCC = 0.66 of all single knockout strains.

- Are there particular categories of functional annotations that are predicted better than others?

Response: The accuracy of predicting across the various gene functions did not show any significant differences, while the accuracy of chromatin factor prediction was the highest, followed by cell cycle regulation (Table R5). We have now included this as a panel in Figure 5.

Table R5 The predictive performance of each function in test set.

Gene function	Accuracy
cell cycle regulation	0.92
chromatin factor	0.94
gene-specific transcription factor	0.87
protein kinase	0.89
ubiquitin(-like) modification	0.89

- Can the authors classify any previously uncharacterised gene into a meaningful functional category using this methodology?

Response: As the machine learning algorithm is designed, with transcriptomic and fluxomic data as input, one can classify any previously uncharacterized gene into the five-class functional classification that was used to train the model, even if the uncharacterized gene has no direct metabolic function. While we demonstrate the feasibility of this approach, and the

suitability to include fluxomics data, the use of this approach to classify uncharacterized genes would largely dependent on the number of classifications. We have now addressed is aspect in lines 329-332 as follows:

While demonstrating the feasibility of this approach and the suitability of yeast-GEM, this framework could gain impact with more knockout transcriptomics, single-function gene annotations, or machine learning approaches that allow for multi-label classification, as would be required to handle e.g. GO term annotations.

Questions about technical details:

- Were 500 features used in each case for flux only, transcriptomics only, flux and transcriptomics?

Response: Following your recommendation, we employed GridSearchCV to fine-tune the features of kPCA, indicating that a selection of 500 features may not universally constitute the optimal choice for every scenario. We have now gathered all such parameter values in Table EV2.

- Which kernel was used in the kPCA?

Response: We have tested the 'linear', 'poly' and 'rbf' kernels. The best kernel varies with the data and machine learning algorithm.

- What criteria were used to optimise the parameters/kernel used?

Response: In the new version of manuscript, we utilize sklearn's GridSearchCV function to simultaneously optimize the parameters of machine learning algorithms and kPCA. The optimization objective of GridSearchCV is to identify the combination of parameters that optimizes model performance by traversing a specified parameter grid. It is primarily used for hyperparameter tuning of the model, with the goal of enhancing the model's predictive accuracy. This is the standard for parameter optimization in the article, we now explicitly refer to GridSearchCV in the Methods section in lines 443-446 as follows:

The parameters of kPCA ('n_components' and 'kernel') as well as those of the Random Forest classifier ('n_estimators', 'max_depth', 'min_samples_split', and 'min_samples_leaf') were optimized using the GridSearchCV function from the scikit-learn library, along with a five-fold cross-validation

- Is the same type of kernel ideal for transforming both the transcriptomics and flux data?

Response: Not necessary. Upon reviewing all the optimal parameters, we generally found that the 'rbf' kernel exhibits the best performance for both the transcriptomics and flux data.

- Why is the data processing pipeline for this ML prediction different from the one used for the single cell GEM fluxes?

Response: We have now systematically evaluated three data preprocessing methods (UMAP, kPCA and no-preprocessing) for single cell classification and gene function prediction. As

kPCA performed best across all datasets and algorithms, we unified the ML data processing pipelines to use kPCA, and have added the updated results in the manuscript.

- Perhaps the authors could comment on considerations or any exploratory data analysis that led them to use UMAP for the single cell data and kPCA for the knockout dataset and why one approach is better than the other for future applications for GEM fluxes as input for ML models.

Response: As detailed in the previous response, we have now evaluated UMAP, kPCA and no-preprocessing and identified kPCA as best performing.

- Limited description of the ML models, their strengths and limitations, the parameters used for each model and any hyperparameter tuning or optimisation that was conducted. Perhaps the authors could adopt the model cards approach (<https://modelcards.withgoogle.com/about>) or a similar documentation strategy being used within the ML community to provide more transparency (at least for their best performing GEM based model).

Response: Based on this and various other comments of the reviewers, we have re-analyzed and extensively rewritten large parts of the machine learning parts of the manuscript. This includes more detailed description of data preprocessing techniques, specific machine learning algorithms used, the process of parameter optimization, as well as the strength and limitations of our approach. In particular, we now explicitly identify the availability of conditions-specific transcriptomic data (which is a prerequisite for each of the demonstrated ML-GEM approaches) as a clear limitation. In essence, each ML simulation necessitates the acquisition of the corresponding transcriptomic profiles. We refer to this in lines 368-370 as follows:

In particular, the ML approaches demonstrated in this manuscript obligate transcriptomics data, thereby highlight a potential limitation of these approaches, as such data is not always readily available.

- How generalisable is the model? Is there any other published transcriptomics dataset from an independent experiment that could be used to conduct cross-validation?

Response: Following your recommendation, we utilized GridSearchCV for executing 5-fold cross-validation, where the dataset underwent a random division into training (70%) and testing (30%) sets, thereby confirming the model's generalizability. These details are now mentioned in lines 443-446.

Culley, Christopher, Supreeta Vijayakumar, Guido Zampieri, and Claudio Angione. 2020. "A Mechanism-Aware and Multiomic Machine-Learning Pipeline Characterizes Yeast Cell Growth." Proceedings of the National Academy of Sciences of the United States of America 117 (31): 18869-79.

Gasch, Audrey P., Feiqiao Brian Yu, James Hose, Leah E. Escalante, Mike Place, Rhonda Bacher, Jad Kanbar, et al. 2017. "Single-Cell RNA Sequencing Reveals Intrinsic and Extrinsic Regulatory Heterogeneity in Yeast Responding to Stress." PLoS Biology 15 (12): e2004050.

Gligorijević, Vladimir, P. Douglas Renfrew, Tomasz Kosciolatek, Julia Koehler Leman, Daniel Berenberg, Tommi Vatanen, Chris Chandler, et al. 2021. "Structure-Based Protein Function Prediction Using Graph Convolutional Networks." *Nature Communications* 12 (1): 3168.

Messner, Christoph B., Vadim Demichev, Julia Muenzner, Simran K. Aulakh, Natalie Barthel, Annika Röhl, Lucía Herrera-Domínguez, et al. 2023. "The Proteomic Landscape of Genome-Wide Genetic Perturbations." *Cell* 186 (9): 2018-34.e21.

Radivojac, Predrag, Wyatt T. Clark, Tal Ronnen Oron, Alexandra M. Schnoes, Tobias Wittkop, Artem Sokolov, Kiley Graim, et al. 2013. "A Large-Scale Evaluation of Computational Protein Function Prediction." *Nature Methods* 10 (3): 221-27.

Reviewer #2:

This manuscript describes the release and test of the newest version of the Yeast GEM, version 9. The subject is timely. The manuscript is very well-written, the study is well designed and conducted, and the obtained results are very interesting.

Response: We are grateful for the constructive feedback from the reviewer.

The following issues should be addressed to improve the quality of the manuscript:

- Line 104 and similar quotes - the authors talk frequently about "condition-specific GEMs (csGEMs)" or "strain-specific GEMs (ssGEMs)"; although I get the idea, that means that we need a different model for every condition, so we need omics data every time we want to change the condition in which we operate. Does that not mean that model alone does not work? At least please discuss the issue.

Response: We indeed do not mean to imply that the model strictly requires omics data, especially when considering "classical" FBA calculations. But we are convinced that the omics-constrained model (if done correctly) provides realistically reduced solution spaces, thereby enhancing their predictive performance. Moreover, the ML approaches that we demonstrate in our manuscript do depend on cs- and ssGEMs. We have now elaborated on this in lines 364-370 as follows:

Thereby, multi-omics analyses and GEM simulations are highly complementary when investigating metabolic regulation. Omics-constrained GEMs reduce the solution space, thereby eliminating biologically infeasible solutions and consequentially resulting in model outcomes with higher confidence. At the same time, this does not mean that it is strictly essential to integrate omics data in Yeast9 before simulations can be performed, just as with any other genome-scale model of metabolism. The ML approaches demonstrated in this manuscript obligate transcriptomics data, thereby highlight a potential limitation of these approaches, as such data is not always readily available.

- Line 173 - should the case described here be solved by the integration of transcriptomics data, given that the impact of redundant gene expression would be accounted for?

Response: This is a good point. In principle, such issues would indeed be addressed by integrating transcriptomics data: an approach that we demonstrate later in the manuscript. We now explicitly mention this in lines 158-159:

This advocates for a more fine-grained consideration of isoenzyme activity, through e.g. the integration of condition-specific transcriptomics data, as demonstrated below.

- Line 199 - describe how relative transcript levels were converted into constraints of the model.

Response:

We make use of the algorithm and MATLAB code that Culley et al. (2020) have previously published. We have now clarified this in lines 507-522 of the Methods section as follows:

The transcriptomic data of single- and double-knockout strains (Kemmeren et al, 2014; O'Duibhir et al, 2014; Sameith et al, 2015) were used to constrain reaction lower and upper bounds, by utilizing the algorithm and MATLAB code previously described by (Culley et al, 2020). Briefly, reaction lower and upper bounds as defined in Yeast9 were defined as representing the wildtype reference strain. Then, non-log-transformed gene expression levels as obtained from microarray experiments (ranging from 117-fold downregulation to 64-fold upregulation in comparison to wildtype) were used to multiply the existing lower and upper bounds of gene-associated reactions. To prevent unrealistic flux bounds, winsorization was applied to smooth extreme values (except for the knockout gene), to make them fit within the 1st and 99th percentile of gene expression values. If a reaction was annotated with multiple genes, the minimum expression level among subunits, and/or the maximum expression level among isoenzymes were used. Through this approach, the original solution space was reshaped to represent the knockout strain specific solution spaces.

- Line 353 - discuss whether the massive number of simulations helped to improve the quality of the underlying model, irrespective of the specific conditions tested.

Response: The reviewer raises a very interesting point. For Yeast9, we have approached the model curation and analysis as two separate steps. First, various curation strategies were employed to improve Yeast8.3.0 towards Yeast9. Then, the developed Yeast9 model was used in large-scale simulations (> 1 million in our manuscripts). Part of these simulations were to directly evaluate the performance of Yeast9 itself (e.g. synthetic lethal simulations), while other simulations aimed to demonstrate the value that Yeast9 brings into multi-omics analyses. However, this curation and analysis pipeline has functioned unidirectional: we did not improve the quality of the underlying model (Yeast9) based on results from the demonstrate applications. As example, we did not investigate the 58,860 + 43,293 false positive and false negative synthetic lethality predictions, to enumerate which of these could be explained by differential gene expression regulation of isoenzymes. We now refer to this in lines 344-346:

However, some limitations still exist for Yeast9. For example, it still lacks high-

resolution details in representing yeast metabolic activities at the organelle level and some reactions for lipid synthesis are not standard. **Our synthetic lethality analysis yielded almost 80% accuracy of prediction, leaving room for improvement.** Thus, further efforts from the yeast community still need to be taken **as part of the circular process** to refine the quality of Yeast9.

Minor issues:

Line 59 - Goffeau's first paper describing the yeast genome seq is dated 1996: Goffeau A, et al. (1996) Life with 6000 genes. *Science* 274(5287):546, 563-7.

Response: Fixed.

Line 83 - Delete "is"

Response: Fixed.

Figure 2e - clarify what was the criteria used to select the tested C and N sources?

Response: In Fig 2, the sources of C/N/P/S were compiled from Biolog experiments ([10.1093/gigascience/giz015](https://doi.org/10.1093/gigascience/giz015)) as have previously been used to evaluate Yeast8 (Lu *et al*, 2019). We now clarify this in lines 147-150:

With a 27% increase in MEMOTE score (Lieven *et al*, 2020), the predictions of single gene essentiality and Biolog-plate measured substrate usage (Kang *et al*, 2019) by Yeast9 were moderately improved compared with Yeast8 (Fig. 2A, 2B).

Reviewer #3:

General comments

The authors present the latest iteration of the consensus yeast genome-scale model (Yeast-GEM 9) and illustrate its utility through various case studies. There have been significant updates since the previous version (Yeast GEM 8). Metabolic modeling of yeast is useful for a broad spectrum of research areas ranging from industrial applications to fundamental studies of metabolic processes and their regulatory mechanisms. The importance of this work is unequivocally evident. However, despite our inclination to recommend this study for publication, there is a major issue concerning its presentation that prevents us from doing so. Starting with the title, the manuscript implies that Yeast GEM 9 has unique capabilities, but fails to show this with a comparison to other yeast models, except for a couple panels in Figure 2. In fact, we believe most demonstrated capabilities are inherent in Yeast GEM 8 (as well as in any good quality model, I should add). Addressing this issue is crucial, either through direct comparisons with Yeast 8 to highlight the advancements or by revising the narrative to reflect the true scope of the model's improvements.

Regardless, there are also several areas of concern that need to be addressed to ensure the study's overall quality (please see major comments below).

Response: We are grateful for the reviewer's kind remarks and detailed comments that we

have used to further improve the manuscript. As the reviewer mentions, there are significant updates between Yeast8 and Yeast9 in many aspects, as we detail in our manuscript, but could also be evidenced by the almost 600 commits on GitHub that separate Yeast8.3.0 from Yeast9.0.0. These changes are a combination of curations of existing model properties, e.g. corrections of reaction reversibility, gene associations and subsystem assignments; but also the introduction of new properties such as thermodynamic information.

Our aim with the manuscript has been two-fold: (a) to publicize the curations and relentless efforts by the yeast community that has resulted in a more accurate, more complete, and more versatile yeast genome-scale model (and by all intentions this work will continue beyond the publication); and (b) to demonstrate that the combination of Yeast9 with integrative analysis of gene expression data can bring novelty to various recently published studies. The reviewer is correct in stating that the applications that we demonstrate could theoretically also be employed with Yeast8.3.0. We even imply this by referring to models in general and not just Yeast9 in the first sentence of the abstract: “Genome-scale metabolic models (GEMs) can facilitate metabolism-focused multi-omics integrative analysis”. We therefore apologize if e.g. the title would have raised other expectations. We however did not deem it valuable to only focus on analyses that would show a different result when comparing the two model versions (albeit we do show the improved performance of Yeast9 in Figure 2, as noted by the reviewer). Instead, we decided to mainly focus on applications that until quite recently would not have been possible due to a lack of data (e.g. single-cell omics; machine learning with large-scale knockout transcriptomics datasets). To refrain from raising expectations that might not be met when reading the paper, we have decided to rephrase the title to avoid the use of the word “Enables”. In addition, we have made edits throughout the manuscript to a similar purpose (e.g. lines 113, 118, 276, 305).

Major Comments (in the order of appearance in the paper):

1) We like the idea of nearly comprehensive deltaGo annotations, which is one example of novel capabilities in Yeast GEM 9. However, I have multiple concerns with the way this addition was handled. Firstly, deltaGo of metabolites are collected from multiple sources. Are these consistent with each other (for the shared metabolites)? Why not use dGpredictor, for example, throughout all metabolites, so that all values are obtained with the same rules? Secondly, the reason some metabolites are missing in the yETFL model seems to be because this resource refused to provide deltaGo values for membrane-associated metabolites, as corrections are needed but unknown for non-aqueous microenvironments. How is this concern addressed here? Thirdly, it is hard to access the deltaGo data (I see it only in .mat model, not in other models like the sbml version, and I also see it in csv files but these are to be searched in github folders), and this data that I found does not indicate the source of the values. And finally, the analysis presented with deltaGos (mapping the values to classical central carbon pathways) is superficial. I thought we could already do this analysis, for example, using MetaCyc pathways and deltaGo values therein. The utility of the deltaGo addition to Yeast GEM is therefore not clear.

Response: We indeed aimed to gather the thermodynamics data from the same source as far as

possible, and we selected yETFL as a reliable source of thermodynamics data gathered for yeast metabolism. For almost 97% of reactions and 96% of metabolites we could derive deltaGo values from yETFL, while an additional 1% of reactions and 2% of metabolites had their deltaGo values derived from dGpredictor and MODELseed, respectively. For the remaining 2% of reactions and metabolites, deltaGo values could not be readily determined, as these are e.g. pseudoreactions or pseudometabolites. We have clarified this in the manuscript in lines 407-415:

The Gibbs free energy change (ΔG°) was added into yeast-GEM. Whenever possible, the ΔG° values for reactions (97%) and metabolites (96%) were gathered from the yETFL model (Oftadeh *et al*, 2021). The ΔG° for reactions that were not contained in the yETFL model were computed by dGPredictor (Wang *et al*, 2021). The ΔG° for metabolites that were not contained in the yETFL model were taken from ModelSEED (Seaver *et al*, 2021).

To assess discrepancies between data from different sources, it was found that errors were around 3% (e.g., H₂O in ModelSEED is -37.54 kcal/mol, compared to -37.21 kcal/mol in yETFL, with an error of 0.8%; reaction r_0006 calculated by dGpredictor is -176.03 kcal/mol, while in yETFL it's -171.22 kcal/mol, with an error of 2.7%). These errors are deemed acceptable.

Regarding "membrane-associated metabolites," not only is it challenging to identify their deltaGo under physiological conditions, but their concentrations are also difficult to determine due to significant variations in local concentrations at the membrane compared to elsewhere in the cytoplasm. In this study, we primarily rely on yETFL to determine deltaGo values and do not take this aspect into further consideration. If such values would become available, they could readily be considered in yeast-GEM as a typical example of community-based curation.

Regarding the accessibility of the deltaGo values from the model, we have purposely not included this in the SBML file, as there neither the SBML L3V2 nor the FBCv3 specifications define how this should be included. While we could devise our own scheme on how to include this (and there would be numerous ways how to do this in the SBML file format), we would deem this to be of limited value, as this information would not be read by most genome-scale modelling tools. In contrast, we deem the YAML file format, which is much less strictly governed and arguably more accessible, as a valuable approach to gather the deltaGo values together with the other model definitions. An important benefit of this file format is that it is very amenable for use in a git-versioned environment, as curations of the model can easily be tracked by comparing differences within this file. The *loadYeastModel* function that is distributed with yeast-GEM would by default load the YAML file. If the SBML file would be given as input, then the deltaGo information is instead read from *data/databases/model_{met|rxn}DeltaG.csv*. We have now clarified in lines 408-410 where the deltaGo values can be found.

The ΔG° values in kJ/mol are available in the YAML version of the model, and when loaded through the provided *loadYeastModel* function, the values are available from the

metDeltaG and rxnDeltaG fields.

2) The result of the synthetic lethality analysis is presented as "Synthetic lethality can be predicted with near-perfect accuracy (Fig. 2c), whereas a few cases of false negative predictions still exist (Supplementary File 2)." This appears to be an overstatement. In fact, since the model predicted no positives, we cannot even talk about recall or precision, because of which I could argue that the model predictions do not have any value. Unless we are missing something, Figure 2c implies that predictions of this model are equivalent to saying "there are no synthetic lethal gene pairs in yeast", which is not very meaningful but is as accurate as the presented predictive algorithm. The authors suggest that the inability to predict positives is partially due to apparent redundancies, which we agree, but then we suggest a different gene deletion algorithm which removes all reactions of a deleted gene no matter if other genes are associated with those reactions and how. How would predictive ability change then? How many of the experimental positives would be predicted?

Response: We are very appreciative of the valuable feedback from this and other reviewers on the synthetic lethality prediction. This forced us to have another critical look at our analysis, which is when we identified a bug in our previous code. We have carefully repeated the analysis, which has yielded different results, which is shown in Table R6 (and Figure 2 in the manuscript). Now, we do have false positive and true positive synthetic lethality predictions (these categories were previously zero), with an overall accuracy of almost 80%.

As we do still observe false negatives (the model predicts genes to be non-lethal, potentially due to redundant isoenzymes, while these might not be expressed in reality), it remains relevant to consider the role of isoenzymes in gene associations. We followed the Reviewer's suggestion and repeated the synthetic lethality analysis with an algorithm that does not consider multiple gene associations. Rather, a reaction will be blocked if it is annotated with the knockout gene, irrespective of the presence of isozymes. This approach yielded drastically worse predictions (Table R6), albeit with less false negative predictions. We now show all these results in Fig. 2, and discuss these results in lines 151-159:

Synthetic lethality can be predicted with almost 80% accuracy (Fig. 2D). Various false negative predictions may be the consequence of reactions that in Yeast9 are annotated with redundant isozymes, which may prevent *in silico* lethality. *In vivo*, however, the expression of the isozymes are transcriptionally regulated (Bradley *et al*, 2019; Zhang *et al*, 2018), which is an aspect that is not considered in GEMs. Worse synthetic lethality predictions were obtained when reactions were disabled even if the knockout gene was an isoenzyme, yielding an accuracy of less than 60%. This advocates for a more fine-grained consideration of isoenzyme activity, through e.g. the integration of condition-specific transcriptomics data, as demonstrated below.

Table R6 Comparison of two gene-deletion algorithms.

	Respecting relationships	AND/OR	Ignore relationships	AND/OR
--	--------------------------	--------	----------------------	--------

tp	6506	18402
tn	419158	296988
fn	43293	31391
fp	58860	181070

3) Flux predictions based on the integration of single-cell data with GIMME algorithm are used to classify cells in two groups. How can we evaluate the predictive power of this classification without a reference point? One possibility is establishing a baseline by doing the same classification with gene expression alone (another possibility is comparing the results with those from Yeast GEM 8, but this general criticism is applicable to pretty much any case study [see above in general comments]). In addition, we do not believe we can yet reach conclusions from this modeling analysis as in "In terms of evolution, heterogeneity at the single-cell level may assist *S. cerevisiae* populations to efficiently adapt to new environments", because the heterogeneity in predicted flux distributions may be an artifact of sparse single-cell data, or any other factor, which may cause inaccurate predictions by the GIMME algorithm. However, some additional steps, such as randomization of cell labels and rerunning the classification *n* times, to establish the statistical significance of predictions may be taken to reach such conclusions.

Response: We are grateful for your constructive suggestions. In response to suggestions from Reviewer 1, we have made various changes in the machine learning approach. We now compare the classification based on fluxomics with transcriptomics, and found that the latter proved to be more conducive to classification under these conditions (Table R1), and discuss this in lines 197-206:

After parameter optimization, the random forest classifier demonstrated 78% accuracy in differentiating single cell sampled from the osmotic stress and unstressed conditions (Fig. 3). The optimized parameters are presented in Table EV2. When the same analysis on the whole transcriptomics dataset yielded 100% accuracy, corroborating the observations by Gasch *et al.* (2017), this implied that the metabolic networks of individual cells were sufficiently different to enable categorizing them by which environment they resided without considering differential expression of non-metabolic gene. However, differential expression of non-metabolic genes are important determinants of the osmotic stress state that likely have strong biological relevance.

Interestingly, when we performed the same comparison in the gene function classification analysis later in the manuscript, we found that fluxomics lead to more accurate classification than transcriptomics.

In the osmostress analysis we now conducted 10 random label classifications, resulting in an average test accuracy of 0.47±0.04. This strengthens our believe that the models generated from single cell transcriptomics with GIMME are representing true biologically relevant mechanisms involved in osmotic stress response, highlighting differences between the two cell types, irrespective of the challenges associated with the sparsity of single-cell data. This is further supported by Figure 3D illustrating that under non-stress conditions, yeast cells predominantly exhibit faster growth compared to stressed cells, even when appreciating for

both biological variability and noise in the data acquisition. Moreover, the heterogeneity observed in Figure 3D might very well play a role in aiding *S. cerevisiae* populations in their adaptation to new environments. We have made significant changes to that section of the manuscript, and have particularly discussed the data sparsity and heterogeneity in lines 188-192 and 212-218:

We used GIMME (Becker & Palsson, 2008) to construct 163 single-cell omics-constrained GEMs (scGEMs) by modify the presence of reactions and metabolites in the model based on transcriptomic data (Fig. 3A). Strictly, due to the nature of single-cell data, it is not known whether the scGEMs reflect true inter-cell variability, or rather only reflect stochasticity in the single-cell data acquisition, as is a challenge in any single-cell approach.

[...]

While generally *S. cerevisiae* grew slower under osmotic stress, a subset of stressed cells grew similarly to those in unstressed cells and vice versa. In this scenario, the alterations in the scGEMs' metabolic network topology, driven by single-cell transcriptomes, result in growth heterogeneity that accurately reflect the true cellular growth state. This shows a possible existence of a resistant phenotype or alternative stress response pathways that might be worth exploring. In terms of evolution, it is possible that heterogeneity of fluxes at the single-cell level might play a role in aiding *S. cerevisiae* populations in their adaptation to new environments.

4) The calculation of the preference score used in the evaluation of nitrogen sources require some clarification and more detailed explanations than what is provided. This method and data obtained from it are counter intuitive. In particular, how can glucose, which has no nitrogen atoms, compensate for the limitation of a nitrogen source?

Response: Here, we added more detailed explanations for this method. The reasoning behind the nitrogen preference score calculations is as follows: during long-term evolution, the yeast's preference for a certain nitrogen source is influenced by the convenience of its utilization. In other words, if nitrogen source A requires more cellular resources for absorption and utilization compared to nitrogen source B, yeast will evolutionarily choose a nitrogen source that is more resource-efficient/energy-efficient and easier to utilize. Or, if possible, the cell might develop more efficient utilization of selected nitrogen sources, through e.g. enzyme evolution, that are in ample supply from its surroundings. Irrespective, a correlation exists between nitrogen source preference and energy-efficiency to utilize that nitrogen source. With glucose as most important energy resource of yeast, its uptake rate is thereby indicative of differences in resource-requirements between nitrogen sources, if all other conditions (such as growth rate) are kept constant. Ammonium nitrogen and glutamine are both direct nitrogen sources, and the cost for yeast to utilize them is relatively low. In contrast, yeast requires more enzymes, energy, and other cellular resources when utilizing isoleucine and phenylalanine compared to the former two. From this perspective, the simulation results align with intuitive understanding and existing literature reports. We have clarified this in lines 239-245:

To check whether yeast-GEM could quantitatively classify the preferred nitrogen source

utilized by *S. cerevisiae*, preference scores for the nitrogen sources ammonium, glutamate, isoleucine and phenylalanine were calculated using Yeast9 (Fig. 4A). During long-term evolutionary processes, the amount of cellular resources devoted to nitrogen uptake and utilization influences *S. cerevisiae* preference for nitrogen sources. *S. cerevisiae* evolutionarily chooses a nitrogen source that is more resource-efficient and easier to utilize. If only limited nitrogen sources are available, *S. cerevisiae* could contrastingly evolve more efficient use of those nitrogen sources. Irrespective, as glucose acts as the main carbon skeleton and energy source for yeast, we defined the nitrogen preference score is the absolute value of the slope between nitrogen and glucose uptake rates, where the uptake of a non-preferred nitrogen source will result in a more significant increase in glucose uptake when compared to a preferred nitrogen source.

5) The authors reach a strong conclusion that the correlation between protein abundance and flux is low, based on fluxes predicted from chemostat data (Figure 4d). Without understanding the uncertainty in the predicted flux values, this conclusion may be premature. Flux variability analysis, or flux sampling as the authors seem to have done, can be used to assess how tight the flux values are, given the experimental constraints. If the variability is generally high, the lack of correlation may also be explained by our low confidence in the predicted flux values, meaning the conclusion is not valid. The related section in Methods "Simplified regulation analysis based on sampled fluxes and absolute proteomics" is unclear and poorly written.

Response: The reviewer is correct that poor correlation can also be due to high variability in one or more of the datasets. However, the variability in the flux sampling did not affect our ρ calculations, as we failed to clarify that we only extracted the mean flux vectors of each simulated condition. This approach provided a non-biased prediction of the feasible flux distribution in that condition, while not considering any measure of the confidence around those predicted flux values. To avoid confusion, we have repeated the analysis, where we have replaced flux sampling with parsimonious FBA and found similar results (most fluxes have a low ρ -value relative to protein abundance). We are therefore comfortable to raise the hypothesis that there may be a low correlation between protein abundance and flux.. For clarity, we have decided to show the pFBA results. The relevant Methods section has been significantly changed in lines 488-504:

Proteomic data and yeast phenotype measurements were collated from literature (Yu et al, 2021a, 2020). Yeast9 was constrained by the measured exchange fluxes, total protein concentrations, and growth rates from aforementioned publications. Then, nitrogen uptake (for which no measurements were available) was minimized and subsequently fixed to the obtained value. Lastly, targeting maximize ATP maintenance, the flux distribution calculated by pFBA (Kochanowski et al, 2021). To determine the protein regulation coefficient ρ , only reactions and proteins that showed nonzero value in a minimum of five out of twelve conditions were considered. The ρ value was determined for each reaction-protein pair individually for each limitation by linear regression between the log-fluxes and log-protein concentrations. When multiple isoenzymes were associated with a reaction, the average regulation coefficient of all corresponding

proteins was computed to determine the final protein regulation coefficient for that reaction. For reactions catalyzed by a complex, the minimal concentration of all subunits was used to compute the protein regulation coefficients.

6) We have several concerns also with the GEM reconstructions for knock-out strains: From the related methods section, where the explanations are not clear, the integration of gene expression data seems to be done by setting flux boundaries as gene expression levels (although how upper and lower fluxes are set differently is also not clear). What is the justification for this? Since the method is also not readily comprehensible in the referred studies, some explanation (or correction in the methods) seems to be necessary. In addition, the referred study from Culley et al. (2020) reports the same level of correlation (0.66) between predicted and measured growth rates as this study found (Fig 5b). Please comment on this. Finally, for a full comparison between transcriptomics-based and fluxomics-based predictions from machine learning, we would need to see the train set/test set accuracy table, as in Table 1, for the transcriptomics-based predictions (Table 1 is for fluxomics-based predictions only). For example, random forests classifier seems like the best predictor with fluxomics in the figure (5c), but is rejected because of poor test set accuracy, which is not shown in the figure but available in the table for fluxomics-based models. There is no such table for transcriptomics-based models.

Response: Regarding the methods of constraining models through transcriptomics, there are currently two main categories. The first involves altering the presence of reactions (Becker & Palsson, 2008), while the second involves changing the upper and lower limits of reactions (Angione & Lió, 2015; Culley et al, 2020; Shlomi et al, 2008). In contrast to enzyme-constrained models, using transcriptomic constraints lacks a theoretical explanation akin to the Michaelis-Menten equation. As the Reviewer correctly interpreted, the gene expression levels were integrated by reshaping the solution space of the model by modifying reaction lower and upper bounds. We have now rephrased this in lines 507-522:

The transcriptomic data of single- and double-knockout strains (Kemmeren *et al*, 2014; O'Duibhir *et al*, 2014; Sameith *et al*, 2015) were used to constrain reaction lower and upper bounds, by utilizing the algorithm and MATLAB code previously described by (Culley *et al*, 2020). Briefly, reaction lower and upper bounds as defined in Yeast9 were defined as representing the wildtype reference strain. Then, non-log-transformed gene expression levels as obtained from microarray experiments (ranging from 117-fold downregulation to 64-fold upregulation in comparison to wildtype) were used to multiply the existing lower and upper bounds of gene-associated reactions. To prevent unrealistic flux bounds, winsorization was applied to smooth extreme values (except for the knockout gene), to make them fit within the 1st and 99th percentile of gene expression values. If a reaction was annotated with multiple genes, the minimum expression level among subunits, and/or the maximum expression level among isoenzymes were used. Through this approach, the original solution space was reshaped to represent the knockout strain specific solution spaces.

The reviewer correctly observes that the single knockout growth predictions had a similar Pearson correlation coefficient (PCC) of 0.66 as the previous analysis by Culley *et al.* (2020). The double knockout growth predictions were however somewhat better (cf. PCC = 0.78 in our analysis and PCC = 0.76 by Culley *et al.* (2020)). The previous analysis used the iSce926 model, which has since been incorporated into Yeast8 and now curated further to Yeast9, demonstrating a gradual improvement of the yeast-GEM model, without (the need of) enormous changes between versions, advocating for its status as a high-quality model. We have discussed this in lines 295-301:

Flux balance analysis (FBA) with growth maximization as the objective function showed a good correlation between the predicted and measured growth rate, with Pearson correlation coefficient (PCC) = 0.66 for single knockout strains and 0.78 for double knockout strains (Fig. 5B). In comparison to Culley *et al.* (2020), which used the pre-Yeast8 *S. cerevisiae* GEM iSce926, which had explicitly been curated for synthetic lethality analysis (Chowdhury *et al.*, 2015) Nonetheless, the relatively moderate improvement in predictive performance is representative of the gradual improvement of yeast-GEM through each newer release.

To facilitate a thorough comparison between transcriptomics-based and fluxomics-based machine learning predictions, we implemented extensive parameter optimization as recommended by Reviewer and have included the new result in the Table R3 and Table 1 in the manuscript to illustrate the disparities.

Minor comments and suggestions:

1) It would be nice to provide a list of modifications in Yeast GEM 9 compared to version 8 (additions and subtractions as reactions, genes, and metabolites) with notes about the reason for each modification.

Response: To provide a full track record of the model curation, we have been making use of version control on GitHub since 2018 (Yeast7.6.0). There, all modifications between each model version can be inspected from different levels. At the highest level, the Releases page (<https://github.com/SysBioChalmers/yeast-GEM/releases>) gives a summary of what changes are made in each model version. In the release notes, the specific Pull Requests are linked, where more specific information can be gathered. One can also compare different release and observe all commits in between: <https://github.com/SysBioChalmers/yeast-GEM/compare/v8.3.0...v9.0.0>. In addition, by providing the model files also in flat-text format enables this view where changes to the model can be directly linked to the pull request of commit that made the change: <https://github.com/SysBioChalmers/yeast-GEM/blame/main/model/yeast-GEM.txt>.

2) As the authors also indicate, deltaGo (standard Gibbs free energy change) values are not sufficient to determine reaction reversibility. Actual deltaG values are also a function of metabolite activity coefficients. Is there a database from which probable ranges of activity for most yeast metabolites can be obtained, so that we can see a range of deltaG (min and max) in addition to deltaGo annotations? A range not crossing 0 would then indicate likely

irreversibility in one direction. This would be very useful and unique.

Response: The YMDB database (<https://www.ymdb.ca/>) provides extensive information on the metabolome of yeast. YMDB contains information of 16040 metabolites in yeast, of which 869 compounds contain concentration data. We wrote code to match these with metabolites in the model, but only found 125 hits (315 when considering metabolites in different compartments). Such low coverage (cf. the 2805 metabolites in Yeast9) precludes systematically assign deltaG ranges to the reactions (keep in mind that concentrations should be known for all substrates and products before the deltaG can be determined).

Irrespective, we did our best with the available data and found some reactions with a deltaG range not crossing 0. As shown in Table R7 and R8, many of these unidirectional reactions tend to involve high-energy compounds like ATP, or form redox power (NADH or NADPH). However, there are very significant caveats in this analysis. Foremost, the lack of readily available quantitative metabolomics data of yeast is striking. We hope that technological advances will result in more of such data being generated in the yeast research community. However, another major issue with such data is that it typically measures intracellular metabolites irrespective of organellar localization. The same metabolite may exist in multiple compartments with large differences in concentrations. A prime example is shown in Table R7 below: ATP synthase has a deltaG range that is always more than 0, which is counterintuitive as this would imply that yeast oxidative phosphorylation runs in reverse. In reality, ATP synthase involves protons in two compartments, where the concentration in the mitochondrial intermembrane space is many fold higher than in the mitochondrial matrix. We therefore do not deem this analysis of such value to include it in the manuscript. Nonetheless, we envision that this could be a topic that might be addressed in future model curations, when more data has become available/accessible. We have included YMDB data, and the code to query YMDB, in the yeast-GEM repository as detailed in <https://github.com/SysBioChalmers/yeast-GEM/pull/364>, and we have elaborated more on the role of metabolite levels on deltaG in the manuscript in lines 168-179:

In attempt to define reference ΔG values, we gathered average metabolite concentrations from the Yeast Metabolome Database (YMDB, Ramirez-Gaona *et al*, 2017), as this would allow the determination of Q . For only 125 of the model metabolites (or 315 when considering compartments) could reported concentrations be found in YMDB. This low coverage precluded a systematic analysis of ΔG values, albeit the ΔG° across metabolism can still be examined. In central carbon metabolism (Figure 2F), pathways such as glycolysis, tricarboxylic acid cycle and pentose phosphate pathway have negative the ΔG° with -22.8 , -9.6 and -13.8 kJ/mol, respectively. The ΔG° of individual reactions, however, can range drastically, implying that metabolites reach concentrations (i.e. Q) that are compatible with a forward flux through these reactions. This exemplifies the importance of considering metabolite concentrations before drawing conclusions on the effect of ΔG° on the functioning of the metabolic network.

Table R7 Reactions with deltaG always less than 0.

r_0033	3',5'-cyclic-nucleotide phosphodiesterase
r_0142	adenosine kinase
r_0147	adenylate cyclase
r_0163	alcohol dehydrogenase (ethanol to acetaldehyde)
r_0193	alpha,alpha-trehalase
r_0194	alpha,alpha-trehalase
r_0211	asparagine synthase (glutamine-hydrolysing)
r_0227	ATPase, cytosolic
r_0445	formate dehydrogenase
r_0491	glycerol-3-phosphate dehydrogenase (NAD)
r_0492	glycerol-3-phosphate dehydrogenase (NAD)
r_0568	inorganic diphosphatase
r_0569	inorganic diphosphatase
r_0713	malate dehydrogenase
r_0714	malate dehydrogenase, cytoplasmic
r_0715	malate dehydrogenase, peroxisomal
r_0718	malic enzyme (NAD)
r_0765	NAD kinase
r_0766	NAD kinase
r_0771	NADH kinase
r_0772	NADH kinase mitochondrial
r_0788	nucleoside diphosphatase
r_0964	reduced glutathione via ABC system
r_1085	V-ATPase, Golgi
r_1086	V-ATPase, vacuole
r_4046	non-growth associated maintenance reaction
r_4264	succinate:NAD ⁺ oxidoreductase

Table R8 Reactions with deltaG always more than 0.

r_0165	mitochondrial alcohol dehydrogenase
r_0962	pyruvate kinase
r_2115	alcohol dehydrogenase, (acetaldehyde to ethanol)
r_4185	oxaloacetate carboxy-lyase (pyruvate-forming)
r_4275	Fe(II):NADP ⁺ oxidoreductase
r_0226	ATP synthase
r_1613	5'-nucleotidase (AMP)
r_0451	fumarase
r_0452	fumarase, cytoplasmic

3)The title of the last section of Results "Growth profiles and gene function can be predicted by Yeast9 constrained with large scale transcriptomic" seems truncated.

Response: We have changed the previously unclear section title to "Large scale transcriptomic empowers yeast-GEM to predict growth profiles and gene function".

4)The data in Table 1 indicates that random forest (RF) classifier was overfitted (test score is much lower than training score which is itself nearly perfect). However, RF models are

especially resilient against overfitting. This raises the question if the modeling parameters are optimized properly for all models. In the RF example, did the authors use out-of-bag scores during model training to adjust the parameters (number of trees, tree depth etc.) appropriately? More broadly, the methodology for ensuring a fair and balanced comparison across the different models used in this analysis needs clarification.

Response: Based on this and other comments from the Reviewers, we have repeated the machine learning analyses by evaluating three methods for data preprocessing: UMAP, KPCA, and no-preprocessing. For each method, we meticulously tuned the model parameters and handled cross-validation utilizing GridSearchCV. However, we were not able to fully resolve the overfitting problem. The issue of overfitting might stem from the complexity of the model, which allows it to perfectly fit every detail of the training data, including noise. This implies that the model may perform well on the training data but poorly when generalized to new data, such as with random forests. This also means that we need to test the fit of different data and models to select the optimal algorithm and parameters. In our now more thorough analysis, we observed in Table R3 that the Naïve Bayes classifier attained the greatest accuracy in predictions and exhibited minimal overfitting. It was particularly noteworthy that the classifier's performance improved markedly when flux data was used, compared to relying solely on transcriptome data. The combination of both data types resulted in an enhanced predictive accuracy for the test set, reaching 90%. Significant changes have been made to the manuscript to address these aspects, as indicated earlier in this letter.

5)Line 391: How is protein localization predicted? I am sure this can be found in previous publications, but it is worth mentioning briefly.

Response: These few lines in the Methods had inadvertently made it in the submitted manuscript, beyond our intention. We have been working on a thorough curation of the protein localization, based on data from the yeast genome database SGD. However, we were not able to make sufficient progress in this aspect to warrant its release into yeast-GEM yet. The relevant text has been removed from the manuscript.

6)Methods section titled "Condition-specific GEMs construction and simulation under nitrogen limitation" is confusing. The first paragraph says the model is constrained with experimental growth rate, but the first step indicates a specific (constant) constraint at 0.1 1/h.

Response: The experimental growth rate in the cited paper is 0.1 h⁻¹. We have made this more clear in lines 451-453:

Yeast9 was constrained by the experimentally measured growth rate and exchange rates (except for nitrogen exchange rate) under nitrogen limitation condition (Yu *et al*, 2021, 2020). The carbohydrate, protein, and RNA ratios in the biomass composition were scaled according to the measured carbohydrate, protein, and RNA abundance in the paper. As a result, 12 phenotype-constrained models were generated.

The preference score of yeast for different nitrogen sources was computed as follows:

- Step 1: Allow for uptake of one nitrogen source at the time, with a fixed growth rate of 0.1 h⁻¹ (i.e. the experimental growth rate).

7)Figure 4c: flux values are hard to visualize. Please sort from large to small (or the other way around) or use small bar graphs instead of small tables.

Response: We have increased the size of the flux values to improve readability.

Reference

- Albe KR, Butler MH & Wright BE (1990) Cellular concentrations of enzymes and their substrates. *Journal of Theoretical Biology* 143: 163–195
- Angione C & Lió P (2015) Predictive analytics of environmental adaptability in multi-omic network models. *Sci Rep* 5: 15147
- Apweiler E, Sameith K, Margaritis T, Brabers N, van de Pasch L, Bakker LV, van Leenen D, Holstege FC & Kemmeren P (2012) Yeast glucose pathways converge on the transcriptional regulation of trehalose biosynthesis. *BMC Genomics* 13: 239
- Becker SA & Palsson BO (2008) Context-Specific Metabolic Networks Are Consistent with Experiments. *PLoS Comput Biol* 4: e1000082
- Bekers KM, Heijnen JJ & van Gulik WM (2015) Determination of the in vivo NAD:NADH ratio in *Saccharomyces cerevisiae* under anaerobic conditions, using alcohol dehydrogenase as sensor reaction. *Yeast* 32: 541–557
- Benschop JJ, Brabers N, van Leenen D, Bakker LV, van Deutekom HWM, van Berkum NL, Apweiler E, Lijnzaad P, Holstege FCP & Kemmeren P (2010) A Consensus of Core Protein Complex Compositions for *Saccharomyces cerevisiae*. *Molecular Cell* 38: 916–928
- Bradley PH, Gibney PA, Botstein D, Troyanskaya OG & Rabinowitz JD (2019) Minor Isozymes Tailor Yeast Metabolism to Carbon Availability. *mSystems* 4: 10.1128/msystems.00170-18
- Cheng C, Wang W-B, Sun M-L, Tang R-Q, Bai L, Alper HS & Zhao X-Q (2022) Deletion of NGG1 in a recombinant *Saccharomyces cerevisiae* improved xylose utilization and affected transcription of genes related to amino acid metabolism. *Front Microbiol* 13
- Chowdhury R, Chowdhury A & Maranas CD (2015) Using Gene Essentiality and Synthetic Lethality Information to Correct Yeast and CHO Cell Genome-Scale Models. *Metabolites* 5: 536–570
- Cui D, Liu L, Zhang X, Lin L, Li X, Cheng T, Wei C, Zhang Y, Zhou Z, Li W, *et al* (2023) Using transcriptomics to reveal the molecular mechanism of higher alcohol metabolism in *Saccharomyces cerevisiae*. *Food Bioscience* 51: 102227
- Culley C, Vijayakumar S, Zampieri G & Angione C (2020) A mechanism-aware and multiomic machine-learning pipeline characterizes yeast cell growth. *Proc Natl Acad*

- Evans C, Bogan KL, Song P, Burant CF, Kennedy RT & Brenner C (2010) NAD⁺ metabolite levels as a function of vitamins and calorie restriction: evidence for different mechanisms of longevity. *BMC Chemical Biology* 10: 2
- Kang K, Bergdahl B, Machado D, Dato L, Han T-L, Li J, Villas-Boas S, Herrgård MJ, Förster J & Panagiotou G (2019) Linking genetic, metabolic, and phenotypic diversity among *Saccharomyces cerevisiae* strains using multi-omics associations. *GigaScience* 8: giz015
- Kemmeren P, Sameith K, van de Pasch LAL, Benschop JJ, Lenstra TL, Margaritis T, O’Duibhir E, Apweiler E, van Wageningen S, Ko CW, *et al* (2014) Large-Scale Genetic Perturbations Reveal Regulatory Networks and an Abundance of Gene-Specific Repressors. *Cell* 157: 740–752
- Lenstra TL, Benschop JJ, Kim T, Schulze JM, Brabers NACH, Margaritis T, van de Pasch LAL, van Heesch SAAC, Brok MO, Groot Koerkamp MJA, *et al* (2011) The Specificity and Topology of Chromatin Interaction Pathways in Yeast. *Molecular Cell* 42: 536–549
- Lieven C, Beber ME, Olivier BG, Bergmann FT, Ataman M, Babaei P, Bartell JA, Blank LM, Chauhan S, Correia K, *et al* (2020) MEMOTE for standardized genome-scale metabolic model testing. *Nat Biotechnol* 38: 272–276
- Lu H, Li F, Sánchez BJ, Zhu Z, Li G, Domenzain I, Marcišauskas S, Anton PM, Lappa D, Lieven C, *et al* (2019) A consensus *S. cerevisiae* metabolic model Yeast8 and its ecosystem for comprehensively probing cellular metabolism. *Nat Commun* 10: 3586
- Makeeva DS, Lando AS, Anisimova A, Egorov AA, Logacheva MD, Penin AA, Andreev DE, Sinitcyn PG, Terenin IM, Shatsky IN, *et al* (2019) Translatome and transcriptome analysis of TMA20 (MCT-1) and TMA64 (eIF2D) knockout yeast strains. *Data in Brief* 23: 103701
- Monk JM, Lloyd CJ, Brunk E, Mih N, Sastry A, King Z, Takeuchi R, Nomura W, Zhang Z, Mori H, *et al* (2017) iML1515, a knowledgebase that computes *Escherichia coli* traits. *Nat Biotechnol* 35: 904–908
- O’Duibhir E, Lijnzaad P, Benschop JJ, Lenstra TL, Leenen D, Groot Koerkamp MJ, Margaritis T, Brok MO, Kemmeren P & Holstege FC (2014) Cell cycle population effects in perturbation studies. *Mol Syst Biol* 10: 732
- Oftadeh O, Salvy P, Masid M, Curvat M, Miskovic L & Hatzimanikatis V (2021) A genome-scale metabolic model of *Saccharomyces cerevisiae* that integrates expression constraints and reaction thermodynamics. *Nat Commun* 12: 4790
- Ramirez-Gaona M, Marcu A, Pon A, Guo AC, Sajed T, Wishart NA, Karu N,

- Djoumbou Feunang Y, Arndt D & Wishart DS (2017) YMDB 2.0: a significantly expanded version of the yeast metabolome database. *Nucleic Acids Research* 45: D440–D445
- Robinson JL, Kocabaş P, Wang H, Cholley P-E, Cook D, Nilsson A, Anton M, Ferreira R, Domenzain I, Billa V, *et al* (2020) An atlas of human metabolism. *Science Signaling* 13: eaaz1482
- Sameith K, Amini S, Groot Koerkamp MJA, van Leenen D, Brok M, Brabers N, Lijnzaad P, van Hooff SR, Benschop JJ, Lenstra TL, *et al* (2015) A high-resolution gene expression atlas of epistasis between gene-specific transcription factors exposes potential mechanisms for genetic interactions. *BMC Biology* 13: 112
- Seaver SMD, Liu F, Zhang Q, Jeffryes J, Faria JP, Edirisinghe JN, Mundy M, Chia N, Noor E, Beber ME, *et al* (2021) The ModelSEED Biochemistry Database for the integration of metabolic annotations and the reconstruction, comparison and analysis of metabolic models for plants, fungi and microbes. *Nucleic Acids Research* 49: D1555
- Shlomi T, Cabili MN, Herrgård MJ, Palsson BØ & Ruppin E (2008) Network-based prediction of human tissue-specific metabolism. *Nat Biotechnol* 26: 1003–1010
- van Wageningen S, Kemmeren P, Lijnzaad P, Margaritis T, Benschop JJ, de Castro IJ, van Leenen D, Groot Koerkamp MJA, Ko CW, Miles AJ, *et al* (2010) Functional Overlap and Regulatory Links Shape Genetic Interactions between Signaling Pathways. *Cell* 143: 991–1004
- Wang L & Hatzimanikatis V (2006) Metabolic engineering under uncertainty--II: analysis of yeast metabolism. *Metab Eng* 8: 142–159
- Wang L, Upadhyay V & Maranas CD (2021) dGPredictor: Automated fragmentation method for metabolic reaction free energy prediction and de novo pathway design. *PLoS Comput Biol* 17: e1009448
- Yu R, Campbell K, Pereira R, Björkeröth J, Qi Q, Vorontsov E, Sihlbom C & Nielsen J (2020) Nitrogen limitation reveals large reserves in metabolic and translational capacities of yeast. *Nat Commun* 11: 1881
- Yu R, Vorontsov E, Sihlbom C & Nielsen J (2021) Non-random organization of flux control mechanisms in yeast central metabolic pathways *Systems Biology*
- Zhang Y, Lin Z, Wang M & Lin H (2018) Selective Usage of Isozymes for Stress Response. *ACS Chem Biol* 13: 3059–3064

14th Jun 2024

Manuscript Number: MSB-2024-12208R

Title: Yeast9: A Yeast Metabolic Model Driving Quantitative Analysis of Metabolism By Integrating Big Data

Dear Dr Kerkhoven,

Thank you again for submitting your revised work to Molecular Systems Biology. We have now heard back from the original three reviewers who evaluated your study. As you will see below, the reviewers are mostly supportive of your manuscript but still have remaining concerns that we would ask you to address in another round of revision. In particular it will be necessary to tone down the conclusions and reword the title, providing a more balanced presentation to the readers. In addition, Reviewer 3 has requested addition of precision-recall analyses and clustering via a dedicated algorithm instead of UMAP.

All other issues raised would need to be satisfactorily addressed. Please let me know in case you would like to discuss in further detail any of the comments, I would be happy to schedule a call.

On a more editorial level, we would ask you to address the following points:

We remind you that we have the following formatting requirements:

1) A .docx formatted version of the manuscript text (including legends for main figures, EV figures and tables). Please make sure that the changes are highlighted to be clearly visible. Alternatively you may choose to submit your manuscript as a LaTeX file.

4) A .docx formatted letter INCLUDING the reviewers' reports and your detailed point-by-point responses to their comments. As part of the EMBO Press transparent editorial process, the point-by-point response is part of the Peer Review File (PRF), which will be published alongside your paper.

5) A complete author checklist, which you can download from our author guidelines (<https://www.embopress.org/page/journal/17574684/authorguide#submissionofrevisions>). Please insert information in the checklist that is also reflected in the manuscript. The completed author checklist will also be part of the PRF.

6) Please note that all corresponding authors are required to supply an ORCID ID for their name upon submission of a revised manuscript.

7) It is mandatory to include a 'Data Availability' section after the Materials and Methods. Before submitting your revision, primary datasets produced in this study need to be deposited in an appropriate public database, and the accession numbers and database listed under 'Data Availability'. Please remember to provide a reviewer password if the datasets are not yet public (see <https://www.embopress.org/page/journal/17574684/authorguide#dataavailability>).

In case you have no data that requires deposition in a public database, please state so in this section. Note that the Data Availability Section is restricted to new primary data that are part of this study. This study includes no data deposited in external repositories.

8) For data quantification: please specify the name of the statistical test used to generate error bars and P values, the number (n) of independent experiments (specify technical or biological replicates) underlying each data point and the test used to calculate p-values in each figure legend. The figure legends should contain a basic description of n, P and the test applied. Graphs must include a description of the bars and the error bars (s.d., s.e.m.). Please provide exact p values.

9) Our journal encourages inclusion of *data citations in the reference list* to directly cite datasets that were re-used and obtained from public databases. Data citations in the article text are distinct from normal bibliographical citations and should directly link to the database records from which the data can be accessed. In the main text, data citations are formatted as follows: "Data ref: Smith et al, 2001" or "Data ref: NCBI Sequence Read Archive PRJNA342805, 2017". In the Reference list,

data citations must be labeled with "[DATASET]". A data reference must provide the database name, accession number/identifiers and a resolvable link to the landing page from which the data can be accessed at the end of the reference. Further instructions are available at .

<https://www.embopress.org/page/journal/17574684/authorguide#expandedview>

11) For more information: There is space at the end of each article to list relevant web links for further consultation by our readers. Could you identify some relevant ones and provide such information as well? Some examples are patient associations, relevant databases, OMIM/proteins/genes links, author's websites, etc...

12) Author contributions: CRediT has replaced the traditional author contributions section because it offers a systematic machine readable author contributions format that allows for more effective research assessment. Please remove the Authors Contributions from the manuscript and use the free text boxes beneath each contributing author's name in our system to add specific details on the author's contribution. More information is available in our guide to authors.

13) Disclosure statement and competing interests: We updated our journal's competing interests policy in January 2022 and request authors to consider both actual and perceived competing interests. Please review the policy <https://www.embopress.org/competing-interests> and update your competing interests if necessary.

14) Every published paper now includes a 'Synopsis' to further enhance discoverability. Synopses are displayed on the journal webpage and are freely accessible to all readers. They include a short stand first (maximum of 300 characters, including space) as well as 2-5 one-sentences bullet points that summarizes the paper. Please write the bullet points to summarize the key NEW findings. They should be designed to be complementary to the abstract - i.e. not repeat the same text. We encourage inclusion of key acronyms and quantitative information (maximum of 30 words / bullet point). Please use the passive voice. Please attach these in a separate file or send them by email, we will incorporate them accordingly.

Please also suggest a striking image or visual abstract to illustrate your article as a PNG file 550 px wide x 300-600 px high. Share synopsis text and image, as well as eTOC:

Please note that these would be the final versions and changes during proofing are usually not allowed

15) As part of the EMBO Publications transparent editorial process initiative (see our policy here:

https://www.embopress.org/transparent-process#Review_Process), Molecular Systems Biology will publish online a Peer Review File (PRF) to accompany accepted manuscripts.

In the event of acceptance, this file will be published in conjunction with your paper and will include the anonymous referee reports, your point-by-point response and all pertinent correspondence relating to the manuscript. Let us know whether you agree with the publication of the PRF and as here, if you want to remove or not any figures from it prior to publication.

Please note that the Authors checklist will be published at the end of the PRF.

Molecular Systems Biology has a "scooping protection" policy, whereby similar findings that are published by others during review or revision are not a criterion for rejection. Should you decide to submit a revised version, I do ask that you get in touch after three months if you have not completed it, to update us on the status.

I look forward to receiving your revised manuscript.

Yours sincerely,

Poonam Bheda, PhD
Scientific Editor
Molecular Systems Biology

Reviewer #1:

I am generally satisfied by the authors' response and changes. Few remaining comments for improving the manuscript quality:

1. I am surprised that only 11 genes are overlapping between GEM and the mutants from the T'omics dataset. Nevertheless, I think it will be important to show results excluding these 11 genes to reduce the circularity.
2. The title is a little too buzz-wordy. Consider making it scientifically accurate and informative about the content of the manuscript

Reviewer #2:

All raised issues were adequately addressed.

Reviewer #3:

The authors have substantially revised their manuscript presenting Yeast 9, the latest genome-scale metabolic model of *Saccharomyces cerevisiae*. They have addressed most of the concerns raised in our initial review. Below, I outline the remaining and newly introduced issues that need attention.

1) Fig. 2D and E, and the associated text:

The authors have revised the table to include predicted positives and have reanalyzed it by disregarding GPR (Gene-Protein-Reaction) associations, as we recommended. However, their focus on accuracy as a metric is problematic. In the context of an imbalanced dataset where the majority of experimental outcomes are non-synthetic lethal (negative), precision and recall are more critical measures than accuracy. Accuracy in this case can be misleading since it is largely influenced by the True Negatives, which do not provide significant insights into the model's predictive capabilities. The authors should compute and discuss the precision and recall to offer a clearer picture of performance. This analysis might also reveal that the difference in predictive performance, with and without considering GPR associations, is minimal. Therefore, completely dismissing the method that ignores GPR connections (as seen in Table 2E) may not be justified.

2) Correlation between protein abundance and flux:

The authors used parsimonious Flux Balance Analysis (pFBA) to estimate flux values and reported a poor correlation with proteomic data. Although I can see how pFBA may be considered an improvement over the previous method, drawing strong conclusions from this analysis concerning the relationship between flux and protein abundance is still premature. pFBA represents only one possible solution within a vast space of potential solutions. Additionally, there is inherent uncertainty in most flux values, which should have been evident from the authors' previous flux sampling analysis. Given that the predicted flux values are not definitive, any poor correlation observed could be a result of these uncertainties. Therefore, the relationship between actual flux values and protein abundance remains uncertain. The authors should moderate their conclusions and discuss these limitations to prevent any misinterpretation by the readers.

3) Although the manuscript has been revised to tone down claims about the novel quantitative capabilities of Yeast 9, the running title ("Yeast9 empowers multi-omics analysis") still implies an enhancement not supported by this paper. As the authors' analysis on knock-out strains show, previous models are also capable of multi-omics analysis and improvements in predictive power are minimal. I also think the revised main title ("Yeast 9: A Yeast Metabolic Model Driving Quantitative Analysis of Metabolism By Integrating Big Data") may still mislead readers into thinking that quantitative modeling is a new feature of yeast genome-scale models. While the title retains some ambiguity that partially addresses this issue, I recommend altering it to more accurately reflect the continuity rather than novelty in the quantitative aspects of the model. This second part of my comment can be taken as just a suggestion.

4) Use of UMAP for Clustering and Visualization (Fig. 3C):

The manuscript indicates that UMAP was used for both clustering and visualization of the data. It is generally acknowledged within the scientific community that while UMAP is a good visualization tool, it is not suitable for clustering due to its focus on preserving local rather than global data structure. For more robust and reliable clustering results, I recommend employing a dedicated clustering algorithm, such as the Leiden algorithm, prior to visualization.

5) In. 490-491: "Yeast9 was constrained by the measured exchange fluxes, total protein concentrations, and growth rates from aforementioned publications."

Which procedure was used to constrain flux with protein concentrations? Please clarify.

6) Several sections of the revised text (e.g., In. 188, 200-201, and 297-299) exhibit grammatical inaccuracies and awkward phrasing. A careful revision of the added text and its integration into the existing manuscript seems necessary.

Reviewer #1:

I am generally satisfied by the authors' response and changes. Few remaining comments for improving the manuscript quality:

1. I am surprised that only 11 genes are overlapping between GEM and the mutants from the Tomics dataset. Nevertheless, I think it will be important to show results excluding these 11 genes to reduce the circularity.

Response: We have revised the manuscript accordingly in line 334-335:

In the test set, the prediction accuracy of chromatin factor is the highest, followed by cell cycle regulation (Fig. 5E). In addition, excluding 11 genes that are included in yeast-GEM from the analysis did not drastically change the performance (Table EV6).

2. The title is a little too buzz-wordy. Consider making it scientifically accurate and informative about the content of the manuscript.

Response: We have rewritten the title to “Yeast9: A Consensus Genome-scale Metabolic Model for *S. cerevisiae* Curated by the Community”.

Reviewer #2:

All raised issues were adequately addressed.

Reviewer #3:

The authors have substantially revised their manuscript presenting Yeast 9, the latest genome-scale metabolic model of *Saccharomyces cerevisiae*. They have addressed most of the concerns raised in our initial review. Below, I outline the remaining and newly introduced issues that need attention.

1) Fig. 2D and E, and the associated text:

The authors have revised the table to include predicted positives and have reanalyzed it by disregarding GPR (Gene-Protein-Reaction) associations, as we recommended. However, their focus on accuracy as a metric is problematic. In the context of an imbalanced dataset where the majority of experimental outcomes are non-synthetic lethal (negative), precision and recall are more critical measures than accuracy. Accuracy in this case can be misleading since it is largely influenced by the True Negatives, which do not provide significant insights into the model's predictive capabilities. The authors should compute and discuss the precision and recall to offer a clearer picture of performance. This analysis might also reveal that the difference in predictive performance, with and without considering GPR associations, is minimal. Therefore, completely dismissing the method that ignores GPR connections (as seen in Table 2E) may not be justified.

Response: We agreed with what you put forward. In the context of the imbalanced dataset, both precision and recall are essential to effectively show the predictive capacity of the model. So, we added this part from Line 158-162:

This advocates for a more fine-grained consideration of isoenzyme activity, through e.g. the integration of condition-specific transcriptomics data, as demonstrated below. It is also worth to mention that the low recall (13.1% in Fig. 2D, 37% in Fig. 2E) and precision (10.0% in Fig. 2D, 9.2% in Fig. 2E) highlight the limitations encountered by GEMs in predicting synthetic lethality.

2) Correlation between protein abundance and flux:

The authors used parsimonious Flux Balance Analysis (pFBA) to estimate flux values and reported a poor correlation with proteomic data. Although I can see how pFBA may be considered an improvement over the previous method, drawing strong conclusions from this analysis concerning the relationship between flux and protein abundance is still premature. pFBA represents only one possible solution within a vast space of potential solutions. Additionally, there is inherent uncertainty in most flux values, which should have been evident from the authors' previous flux sampling analysis. Given that the predicted flux values are not definitive, any poor correlation observed could be a result of these uncertainties. Therefore, the relationship between actual flux values and protein abundance remains uncertain. The authors should moderate their conclusions and discuss these limitations to prevent any misinterpretation by the readers.

Response: We appreciate the reviewer's concern regarding the use of single pFBA-derived flux distributions to make definitive statements about correlations between fluxomes and proteomes. While we had previously replaced the flux sampling analysis with the pFBA-based analysis as a response to a comment made by this same reviewer in a previous round of review, we are now convinced that the reader is best served with being provided both sets of results. We have therefore amended Table EV4, and have now explicitly discussed this from Line 275 to 286:

Here, $\rho \approx 1$ signifies that changes in simulated fluxes can largely be explained by protein concentration changes. When based on fluxes predicted from parsimonious FBA (pFBA), most reactions from carbohydrate metabolism and amino acid metabolism exhibited weak protein regulation coefficients (Fig. 4D), with only 1.7% reactions revealing high coefficients ($\rho > 0.5$, Table EV4). As pFBA only yields single flux distributions, ρ values were also determined based on mean fluxes from random sampling of the solution space, which yielded similar results (Table EV4). Both analyses imply a low correlation between protein abundance and flux, albeit it cannot be excluded that uncertainty of the predicted fluxes obscure the true correlation between protein abundance and flux.

Regardless, the previous analysis implies that the direction of change might be more distinctly conserved. Additional post-translational regulatory mechanisms might be implicated in this discrepancy.

3) Although the manuscript has been revised to tone down claims about the novel quantitative capabilities of Yeast 9, the running title ("Yeast9 empowers multi-omics analysis") still implies an enhancement not supported by this paper. As the authors' analysis on knock-out strains show, previous models are also capable of multi-omics analysis and improvements in predictive power are minimal. I also think the revised main title ("Yeast 9: A Yeast Metabolic Model Driving Quantitative Analysis of Metabolism By Integrating Big Data") may still mislead readers into thinking that quantitative modeling is a new feature of yeast genome-scale models. While the title retains some ambiguity that partially addresses this issue, I recommend altering it to more accurately reflect the continuity rather than novelty in the quantitative aspects of the model. This second part of my comment can be taken as just a suggestion.

Response: As suggested, we have revised the title and running title:

Running title: Yeast9: new version of *S. cerevisiae* GEM

Main title: Yeast9: A Consensus Genome-scale Metabolic Model for *S. cerevisiae*
Curated by the Community

4) Use of UMAP for Clustering and Visualization (Fig. 3C):

The manuscript indicates that UMAP was used for both clustering and visualization of the data. It is generally acknowledged within the scientific community that while UMAP is a good visualization tool, it is not suitable for clustering due to its focus on preserving local rather than global data structure. For more robust and reliable clustering results, I recommend employing a dedicated clustering algorithm, such as the Leiden algorithm, prior to visualization.

Response: For transcriptomic data (such as single-cell RNA sequencing data), we can use the Leiden algorithm or the Hierarchical algorithm (Shalek *et al*, 2013) for cell classification, but it is also acceptable to use UMAP directly (Yang *et al*, 2021). It seems that Leiden algorithm is not suitable to process flux data for cell classification (Figure R1). In contrast, the Hierarchical method achieves a 70% accuracy rate (Figure R2). Considering the high consistency of the Hierarchical method with UMAP, the original images and descriptions in the text seem appropriate. We have therefore decided not to modify the manuscript in this aspect.

Figure R1 Leiden Clustering with UMAP Visualization. Each color indicates a cell category grouped by Leiden algorithm.

Figure R2 Hierarchical Clustering with UMAP Visualization. Each color indicates a cell category grouped by Hierarchical algorithm. Please note, the color represents the classification results from the Hierarchical algorithm. The actual classification can be found in Fig. 3C in the main text.

5) ln. 490-491: "Yeast9 was constrained by the measured exchange fluxes, total protein concentrations, and growth rates from aforementioned publications." Which procedure was used to constrain flux with protein concentrations? Please clarify.

Response: The models were constrained by the total cellular protein concentrations, not individual protein concentrations (this would have biased the analysis of protein regulation coefficients. The total cellular protein concentration of the biomass equation was modified with the "scaleBioMass" function from the yeast-GEM repository. To avoid misunderstanding, we have revised the text from line 501 to 503:

Yeast9 was constrained by the measured exchange fluxes and growth rates. The biomass composition was altered according to total protein concentrations from aforementioned publications, using the "scaleBioMass" function in the yeast-GEM repository. Then,

nitrogen uptake (for which no measurements were available) was minimized and subsequently fixed to the obtained value.

6) Several sections of the revised text (e.g., ln. 188, 200-201, and 297-299) exhibit grammatical inaccuracies and awkward phrasing. A careful revision of the added text and its integration into the existing manuscript seems necessary.

Response: We have made various grammatical changes and rephrased sentences throughout the manuscript.

Ref.

Shalek AK, Satija R, Adiconis X, Gertner RS, Gaublomme JT, Raychowdhury R, Schwartz S, Yosef N, Malboeuf C, Lu D, *et al* (2013) Single-cell transcriptomics reveals bimodality in expression and splicing in immune cells. *Nature* 498: 236–240

Yang Y, Sun H, Zhang Y, Zhang T, Gong J, Wei Y, Duan Y-G, Shu M, Yang Y, Wu D, *et al* (2021) Dimensionality reduction by UMAP reinforces sample heterogeneity analysis in bulk transcriptomic data. *Cell Reports* 36: 109442

Yu R, Campbell K, Pereira R, Björkeröth J, Qi Q, Vorontsov E, Sihlbom C & Nielsen J (2020) Nitrogen limitation reveals large reserves in metabolic and translational capacities of yeast. *Nat Commun* 11: 1881

Yu R, Vorontsov E, Sihlbom C & Nielsen J (2021) Non-random organization of flux control mechanisms in yeast central metabolic pathways *Systems Biology*

8th Jul 2024

Manuscript Number: MSB-2024-12208RR

Title: Yeast9: A Consensus Genome-scale Metabolic Model for *S. cerevisiae* Curated by the Community

Dear Dr Kerkhoven,

Thank you for the submission of your revised manuscript to Molecular Systems Biology. I am pleased to inform you that we will be able to accept your manuscript pending the following final amendments:

1) Please format the Data availability section describing how the data, code etc. have been made available according to the example below:

"The datasets and computer code produced in this study are available in the following databases:

- Chip-Seq data: Gene Expression Omnibus GSE46748 (<https://www.ncbi.nlm.nih.gov/geo/query/acc.cgi?acc=GSE46748>)
- Modeling computer scripts: GitHub (<https://github.com/SysBioChalmers/GECKO/releases/tag/v1.0>)
- [data type]: [full name of the resource] [accession number/identifier] ([doi or URL or identifiers.org/DATABASE:ACCESSION])"

2) Author contributions: Please remove it from the manuscript and specify author contributions in our submission system. CRediT has replaced the traditional author contributions section because it offers a systematic machine-readable author contributions format that allows for more effective research assessment. You are encouraged to use the free text boxes beneath each contributing author's name to add specific details on the author's contribution. More information is available in our guide to authors:

<https://www.embopress.org/page/journal/17574684/authorguide#authorshipguidelines>

3) Our journal encourages inclusion of *data citations in the reference list* to directly cite datasets that were re-used and obtained from public databases. Data citations in the article text are distinct from normal bibliographical citations and should directly link to the database records from which the data can be accessed. In the main text, data citations are formatted as follows: "Data ref: Smith et al, 2001" or "Data ref: NCBI Sequence Read Archive PRJNA342805, 2017". In the Reference list, data citations must be labeled with "[DATASET]". A data reference must provide the database name, accession number/identifiers and a resolvable link to the landing page from which the data can be accessed at the end of the reference. Further instructions are available at .

4) All materials and methods need to be described in the main text using our 'Structured Methods' format, which is required for all research articles. According to this format, the Methods section includes a Reagents and Tools Table (listing key reagents, experimental models, software and relevant equipment and including their sources and relevant identifiers) followed by a Methods and Protocols section describing the methods using a step-by-step protocol format. The aim is to facilitate adoption of the methodologies across labs. Please remove the Reagents and Tools table from the main manuscript and upload it as a separate file according to our latest template.

More information on how to adhere to this format as well as a downloadable template (.docx) for the Reagents and Tools Table can be found in our author guidelines: <https://www.embopress.org/page/journal/17574684/authorguide#structuredmethods>

5) Please rename the 'Relevant web links' section of the manuscript to 'For more information'

6) Please place individual sections of the manuscript in the following order: Title page - Abstract & Keywords - Introduction - Results - Discussion - Methods - Data Availability - Acknowledgements - Disclosure and Competing Interests Statement - For More Information - References - Figure Legends - Expanded View Figure Legends.

7) For the figures and figure legends, please take care of the following:

- Please note that the exact p values are required. These are not provided in the figure or legends of figures 3d-e; 5b-c.
- Please indicate the statistical test used for data analysis in the legend of figure 5c.
- Please note that the box plots need to be defined in terms of minima, maxima, centre, bounds of box and whiskers, and percentile in the legend of figure 3e.
- Please note that information related to n is missing in the legend of figure 3e.

8) Tables: Tables EV1 and EV3 should be renamed to Dataset EV3 and Dataset EV4, respectively. Their source file names, titles in our manuscript submission system, sheet names in the Excel files, and callouts in the manuscript all need to be updated accordingly; the remaining EV Tables then need to be re-numbered and their names/titles/callouts corrected accordingly. The legends for all EV Tables and Datasets should be inserted as separate tabs in each corresponding Excel file, not uploaded as a separate file.

9) Synopsis:

- Synopsis text file: Please remove the synopsis image from the synopsis text file. Please also tone down the language of the synopsis text standfirst ("Integrated with big data...").

- Please check your synopsis text and image before submission with your revised manuscript. Please be aware that in the proof

stage minor corrections only are allowed (e.g., typos).

10) As part of the EMBO Publications transparent editorial process initiative (see our policy here: https://www.embopress.org/transparent-process#Review_Process), Molecular Systems Biology will publish online a Peer Review File (PRF) to accompany accepted manuscripts. This file will be published in conjunction with your paper and will include the anonymous referee reports, your point-by-point response and all pertinent correspondence relating to the manuscript. Let us know whether you agree with the publication of the PRF and as here, if you want to remove or not any figures from it prior to publication. Please note that the Authors checklist will be published at the end of the PRF.

11) Please provide a point-by-point letter INCLUDING my comments as well as the reviewer's reports and your detailed responses (as Word file).

I look forward to reading a new revised version of your manuscript as soon as possible.

Yours sincerely,

Poonam Bheda, PhD
Scientific Editor
Molecular Systems Biology

Editor:

1) Please format the Data availability section describing how the data, code etc. have been made available according to the example below:

"The datasets and computer code produced in this study are available in the following databases:

- Chip-Seq data: Gene Expression Omnibus GSE46748

(<https://www.ncbi.nlm.nih.gov/geo/query/acc.cgi?acc=GSE46748>)

- Modeling computer scripts: GitHub

(<https://github.com/SysBioChalmers/GECKO/releases/tag/v1.0>)

- [data type]: [full name of the resource] [accession number/identifier] ([doi or URL or identifiers.org/DATABASE:ACCESSION])"

Response: We have modified the Data Availability section accordingly.

2) Author contributions: Please remove it from the manuscript and specify author contributions in our submission system. CRediT has replaced the traditional author contributions section because it offers a systematic machine-readable author contributions format that allows for more effective research assessment. You are encouraged to use the free text boxes beneath each contributing author's name to add specific details on the author's contribution. More information is available in our guide to authors:

<https://www.embopress.org/page/journal/17574684/authorguide#authorshipguidelines>

Response: We have moved the Author contributions from the manuscript to the manuscript submission system.

3) Our journal encourages inclusion of *data citations in the reference list* to directly cite datasets that were re-used and obtained from public databases. Data citations in the article text are distinct from normal bibliographical citations and should directly link to the database records from which the data can be accessed. In the main text, data citations are formatted as follows: "Data ref: Smith et al, 2001" or "Data ref: NCBI Sequence Read Archive PRJNA342805, 2017". In the Reference list, data citations must be labeled with "[DATASET]". A data reference must provide the database name, accession number/identifiers and a resolvable link to the landing page from which the data can be accessed at the end of the reference. Further instructions are available at

<https://www.embopress.org/page/journal/17574684/authorguide#referencesformat>.

Response: We have now included five data citations in the reference list.

4) All materials and methods need to be described in the main text using our 'Structured Methods' format, which is required for all research articles. According to this format, the Methods section includes a Reagents and Tools Table (listing key reagents, experimental

models, software and relevant equipment and including their sources and relevant identifiers) followed by a Methods and Protocols section describing the methods using a step-by-step protocol format. The aim is to facilitate adoption of the methodologies across labs. Please remove the Reagents and Tools table from the main manuscript and upload it as a separate file according to our latest template.

More information on how to adhere to this format as well as a downloadable template (.docx) for the Reagents and Tools Table can be found in our author guidelines:

<https://www.embopress.org/page/journal/17574684/authorguide#structuredmethods>

Response: To the best of our judgement, we have ensured to comply with the Structured Methods guidelines, and we have provided the Reagents and Tools Table as a separate file.

5) Please rename the 'Relevant web links' section of the manuscript to 'For more information'

Response: We have made the request change.

6) Please place individual sections of the manuscript in the following order: Title page - Abstract & Keywords - Introduction - Results - Discussion - Methods - Data Availability - Acknowledgements - Disclosure and Competing Interests Statement - For More Information - References - Figure Legends - Expanded View Figure Legends.

Response: We have ensured that the manuscript file follows the instructions.

7) For the figures and figure legends, please take care of the following:

- Please note that the exact p values are required. These are not provided in the figure or legends of figures 3d-e; 5b-c.
- Please indicate the statistical test used for data analysis in the legend of figure 5c.
- Please note that the box plots need to be defined in terms of minima, maxima, centre, bounds of box and whiskers, and percentile in the legend of figure 3e.
- Please note that information related to n is missing in the legend of figure 3e.

Response: We have now explicitly included the *p*-values and summary statistics in Figures 3d, 5b-c, and clarified the legends of Figures 3d-e. For Figure 3e, we refer in the figure legend to Dataset EV3, as including all *p*-values in either the figure or the legend would not have benefitted the formatting of the paper.

8) Tables: Tables EV1 and EV3 should be renamed to Dataset EV3 and Dataset EV4, respectively. Their source file names, titles in our manuscript submission system, sheet names in the Excel files, and callouts in the manuscript all need to be updated accordingly; the

remaining EV Tables then need to be re-numbered and their names/titles/callouts corrected accordingly. The legends for all EV Tables and Datasets should be inserted as separate tabs in each corresponding Excel file, not uploaded as a separate file.

Response: We have renamed the respective tables to datasets, updated all references to them, and have moved the table and dataset legends into their respective Excel files.

9) Synopsis:

- Synopsis text file: Please remove the synopsis image from the synopsis text file. Please also tone down the language of the synopsis text standfirst ("Integrated with big data...").
- Please check your synopsis text and image before submission with your revised manuscript. Please be aware that in the proof stage minor corrections only are allowed (e.g., typos).

Response: We have edited the synopsis to tone down the language, in agreement with earlier comments raised by the reviewers regarding the manuscript title.

10) As part of the EMBO Publications transparent editorial process initiative (see our policy here: https://www.embopress.org/transparent-process#Review_Process), Molecular Systems Biology will publish online a Peer Review File (PRF) to accompany accepted manuscripts. This file will be published in conjunction with your paper and will include the anonymous referee reports, your point-by-point response and all pertinent correspondence relating to the manuscript. Let us know whether you agree with the publication of the PRF and as here, if you want to remove or not any figures from it prior to publication. Please note that the Authors checklist will be published at the end of the PRF.

Response: We agree with publication of the PRF and do not request for any changes to the earlier communications.

11) Please provide a point-by-point letter INCLUDING my comments as well as the reviewer's reports and your detailed responses (as Word file).

Response: We have hereby documented our responses to the editor's comments, and highlighted the changes in the manuscript.

31st Jul 2024

Manuscript number: MSB-2024-12208RRR

Title: Yeast9: A Consensus Genome-scale Metabolic Model for *S. cerevisiae* Curated by the Community

Dear Dr Kerkhoven,

Thank you again for sending us your revised manuscript. We are now satisfied with the modifications made and I am pleased to inform you that your paper has been accepted for publication.

Yours sincerely,

Sincerely,

Poonam Bheda, PhD
Scientific Editor
Molecular Systems Biology
